# Mapping human tissues with highly multiplexed RNA in situ hybridization

Kian Kalhor [1,13], Chien-Ju Chen[1,2,13], Ho Suk Lee[1,3], Matthew Cai[1], Mahsa Nafisi [1], Richard Que[1], Carter R. Palmer[4,5], Yixu Yuan[1], Yida Zhang[6], Xuwen Li [7], Jinghui Song[1], Amanda Knoten[8], Blue B. Lake[1,7], Joseph P. Gaut[9], C. Dirk Keene [10], Ed Lein [11], Peter V. Kharchenko[6,7], Jerold Chun [4], Sanjay Jain [8,9], Jian-Bing Fan[12] & Kun Zhang [1,7] ✉

In situ transcriptomic techniques promise a holistic view of tissue organization and cell-cell interactions. There has been a surge of multiplexed RNA in situ mapping techniques but their application to human tissues has been limited due to their large size, general lower tissue quality and high autofluorescence. Here we report DART-FISH, a padlock probe-based technology capable of profiling hundreds to thousands of genes in centimeter-sized human tissue sections. We introduce an omni-cell type cytoplasmic stain that substantially improves the segmentation of cell bodies. Our enzyme-free isothermal decoding procedure allows us to image 121 genes in large sections from the human neocortex in <10 h. We successfully recapitulated the cytoarchitecture of 20 neuronal and non-neuronal subclasses. We further performed in situ mapping of 300 genes on a diseased human kidney, profiled >20 healthy and pathological cell states, and identified diseased niches enriched in transcriptionally altered epithelial cells and myofibroblasts.

Analyzing single-cell expression of genes in their spatial context plays a critical role in deciphering the complex cellular organization in multicellular organisms[1,2]. Gene expression in its spatial context is especially important in fields such as embryo development[3], neuroscience[4], and in histopathology[5]. The emergence of single-molecule fluorescence in situ hybridization (smFISH, Supplementary Table 1 for all acronyms in the manuscript) methods allowed us to simultaneously measure several RNA species in single cells[6,7] by imaging fluorophore-tagged DNA oligos, or probes, that tile the RNA molecules. Because of its high sensitivity, smFISH has become the gold standard assay to measure RNA expression in situ and has been used to show the importance of RNA localization in cell migration, neuron connectivity,

and local protein synthesis[8,9]. However, since smFISH is limited by spectral overlap of the fluorophores, it has limited multiplexing capacity[10], and does not scale well for tasks such as resolving cellular heterogeneity in complex tissues, which require profiling hundreds of RNA species.

Recently, in situ hybridization techniques with combinatorial encoding have emerged in which the identity of hundreds or thousands of RNA species can be decoded with tens of FISH cycles[11,12]. Although these methods have increased the multiplexity by 2-3 orders of magnitude compared to smFISH, they typically require longer target RNA transcripts (>1.5kb), restricting the analysis of important molecules such as neuropeptides and interferons[11,13]. Furthermore, because

[1]Department of Bioengineering, University of California San Diego, La Jolla, CA, USA. [2]Program in Bioinformatics and Systems Biology, University of California San Diego, La Jolla, CA, USA. [3]Department of Electrical Engineering, University of California San Diego, La Jolla, CA, USA. [4]Sanford Burnham Prebys Medical Discovery Institute, La Jolla, CA, USA. [5]Program in Biomedical Sciences, University of California San Diego, La Jolla, CA, USA. [6]Department of Biomedical Informatics, Harvard Medical School, Boston, MA, USA. [7]Altos Labs, San Diego, CA, USA. [8]Department of Medicine, Washington University School of Medicine, St. Louis, MO, USA. [9]Department of Pathology and Immunology, Washington University School of Medicine, St.Louis, MO, USA. [10]Department of Laboratory Medicine and Pathology, University of Washington School of Medicine, Seattle, WA, USA. [11]Allen Institute for Brain Science, Seattle, WA 98103, USA. [12]Illumina, San Diego, CA, USA. [13]These authors contributed equally: Kian Kalhor, Chien-Ju Chen. ✉e-mail: kzhang@bioeng.ucsd.edu

of the low signal-to-noise ratio (SNR) from detected transcripts, these methods need high magnification objectives with high numerical aperture (NA), making it difficult and time-consuming to image large regions of interest (ROIs). The low SNR also makes it challenging to apply these methods to human tissues which may have a high auto-fluorescence background caused by lipofuscin granules[14,15], proteins such as collagen and elastin[16], or mitochondria[17,18]. Methods that ligate padlock probes annealing to mRNA derivatives, followed by rolling circle amplification (RCA) have been employed to boost SNR from individual transcripts. However, these methods are associated with high probe set expenses and complex decoding procedures. They further lack an efficient approach to stain the cell bodies for segmentation[19–21] (Supplementary Table 2).

With the advent of sequencing-based spatial transcriptomics methods[22–27], transcriptome-wide profiling of RNA molecules in tissue sections was made possible by transferring the RNA molecules to a slide coated with spatially-barcoded oligos. In this way, the spatial information of each RNA molecule can be registered through next-generation sequencing. Nevertheless, when compared to in situ methods, sequencing-based spatial transcriptomic tools in general have lower capture efficiency, complex slide preparation procedures, higher sequencing costs, and limited spatial resolution due to feature size and lateral diffusion[28].

Here, we developed Decoding Amplified taRgeted Transcripts with Fluorescence in situ Hybridization (DART-FISH) to overcome some of these limitations. The key technical features include a robust barcoding scheme, a set of molecular protocols for padlock probe production in large pools, in situ padlock capture and amplification, a cytoplasmic stain called RiboSoma, isothermal and enzyme-free decoding, and a computational method for decoding features at the pixel level from dense fluorescent images based on sparse deconvolution. We benchmarked DART-FISH by measuring 121 genes in a large section (~30 mm²) of the human primary motor cortex (M1C). We validated its sensitivity and specificity by comparing it to RNAscope, a commercially available smFISH method (Methods). Moreover, we successfully recapitulated the spatial organization of major neuronal and non-neuronal cell types, detected short neuropeptide genes (e.g., *SST* and *NPY*), and validated a deep layer neuron marker (*TMSB10*). Finally, we applied DART-FISH to measure 300 genes in a diseased human kidney section and characterized the spatial distribution of normal and disease-altered cell types and pathological niches. Overall, the DART-FISH workflow provides solutions to several foundational problems in the field while remains easy to implement and requires no specialized or custom-made equipment.

## Results

### DART-FISH framework

DART-FISH involves in situ feature generation by padlock probe capture of targeted transcripts and rolling circle amplification (RCA), followed by a highly robust decoding process of sequential isothermal hybridization. (Fig. 1a, Methods). Specifically, RNA molecules in fresh-frozen tissue sections are fixed with paraformaldehyde (PFA), permeabilized, and then reverse-transcribed with a mixture of random and poly-deoxythymidine (dT) primers. To assess the RNA content in human tissues as well as the retention of the cDNA molecules in situ, we added a 5' handle to the reverse-transcription primers to enable the collective visualization of all cDNA molecules with fluorescent oligos (Fig. 1b). We call this labeling method RiboSoma because the resulting signal labels the cell bodies. During protocol optimization, we noticed that crosslinking the cDNA molecules immediately after reverse-transcription to a polyacrylamide (PA) gel enhances the RiboSoma signal (Supplementary Fig. 1a) suggesting better retention of cDNA in situ throughout the DART-FISH protocol. This cDNA embedding strategy also led to 1.5-fold median increase of the feature count per gene (Supplementary Fig. 1b, c), compared to when the

polyacrylamide gel is cast after RCA. Thus, RiboSoma serves as a marker for cDNA content of the tissue and provides a quality control for in situ reactions.

Following gel embedding and RNA digestion, cDNA molecules are hybridized with a library of padlock probes and circularized at a high temperature to ensure specificity[29,30]. On their backbone, padlock probes carry a universal sequence used for amplification and gene-specific barcodes. The circularized padlock probes are then rolling-circle-amplified, generating RCA colonies in situ (rolonies) with hundreds of copies of barcode sequences concatenated in the form of a DNA nanoball. The rolonies are then covalently attached to the poly-acrylamide gel to secure their positions during decoding. The result of the experiment is then assessed in the "anchor round" imaging, where fluorescent probes are hybridized to the universal sequences and the 5' handles on cDNA molecules to visualize the spatial distribution of all rolonies and cells (i.e., RiboSoma, Fig. 1b).

To achieve high multiplexity within only a few rounds of imaging, combinatorial labeling was used to generate gene-specific barcodes[31]. In this barcoding scheme, $n$ rounds of imaging are performed where every barcode is "on" in exactly $k$ rounds and "off" in other rounds (Fig. 1c). When "on", the barcode signals in one of the three fluorescent channels; it emits no fluorescence when "off". With $n$ rounds of imaging, a total of $\binom{n}{k}3^k$ unique barcodes can be generated, allowing us to measure hundreds of RNA species with limited rounds of decoding ($n = 6$ and $k = 3$ in Fig. 1 with 540 valid barcodes). This can be extended to 7 rounds of decoding for up to 945 genes ($k = 3$), 8 rounds of decoding for 5670 genes ($k = 4$), and so on. This barcoding scheme has a proven robustness evident by its wide adoption by Illumina's gene expression, SNP genotyping and DNA methylation arrays[31–33]. Hence, DART-FISH uses a barcoding strategy that can theoretically generate enough diversity to encode hundreds to thousands of genes within less than 10 rounds of imaging.

To implement this barcoding system such that the decoding process is fast and robust, gene-specific barcodes are created by the concatenation of $k$ 20-nucleotide-long decoder sequences placed on the backbone of padlock probes[34]. The decoder sequences are derived from Illumina BeadArray technology and have limited cross-hybridization[31] (Supplementary Data 1). In each round of imaging, three unique fluorescent decoding probes are hybridized and imaged. Rolonies will be "on" only if a decoding probe that corresponds to one of their decoder sequences is present. After imaging, the decoding probes are stripped and washed away at room temperature to prepare for the next round (Fig. 1b and e). During this procedure, the rolonies are stable with minimal movement, degradation and background buildup (Supplementary Fig. 1d). Note that this process enables rapid and reliable decoding since it depends solely on the hybridization of short oligonucleotides at room temperature, eliminating the need for sophisticated temperature control setups and avoiding the complications of performing enzymatic reactions on a microscope. Thus, DART-FISH uses an enzyme-free and isothermal method to decode the rolonies which allows short between-cycle preparation times.

It has been shown that increasing the number of padlock probes per gene leads to a higher detection sensitivity in situ[35]. For such applications, it is common to pool individually synthesized padlock probes[35–37]. This strategy, while manageable for small-scale studies, would be prohibitively expensive when probing hundreds of genes is desired. To overcome this limitation, we adapted an enzymatic protocol to produce thousands of padlock probes in-house starting from an oligo pool synthesized on microarrays[38] (Methods, Supplementary Fig. 3c). We were able to target 121 genes each with up to 50 padlock probes for less than 25% the cost of the direct synthesis option. Note in our strategy the cost per probe decreases further by including more probes in the pool, whereas for direct synthesis the cost per probe remains constant. Consequently, individually synthesizing 20,000 probes to target 400 genes is almost 10 times as expensive as array

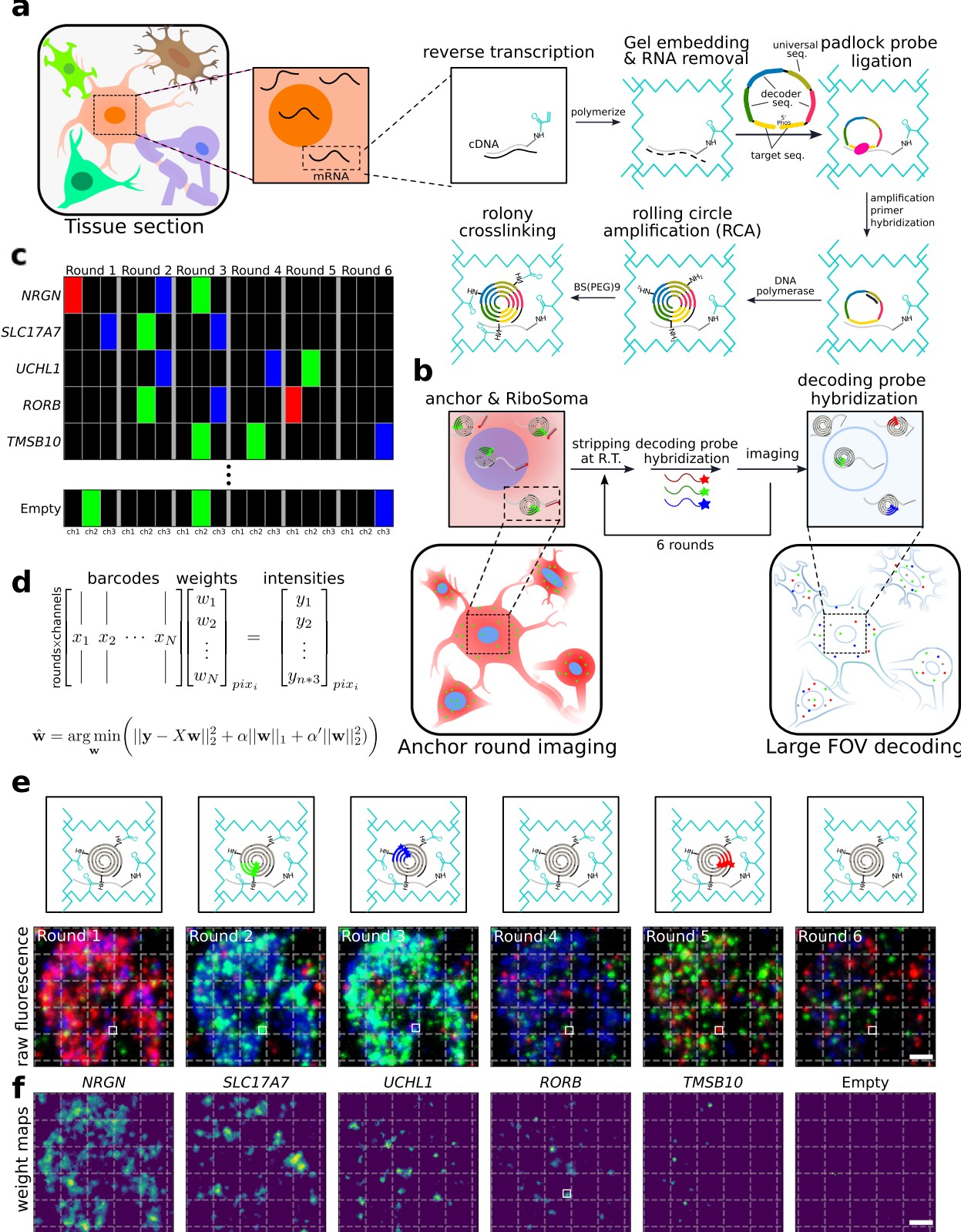

**d**

$$\begin{matrix} \text{barcodes} & \text{weights} & \text{intensities} \end{matrix}$$

$$\begin{bmatrix} | & | & & | \\ x_1 & x_2 & \cdots & x_N \\ | & | & & | \end{bmatrix} \begin{bmatrix} w_1 \\ w_2 \\ \vdots \\ w_N \end{bmatrix}_{pix_i} = \begin{bmatrix} y_1 \\ y_2 \\ \vdots \\ y_{n*3} \end{bmatrix}_{pix_i}$$

$$\hat{\mathbf{w}} = \arg\min_{\mathbf{w}} \left( ||\mathbf{y} - X\mathbf{w}||_2^2 + \alpha ||\mathbf{w}||_1 + \alpha' ||\mathbf{w}||_2^2 \right)$$

synthesis. To fully utilize this feature, multiple probe sets that, for instance, target different organs or organisms can be pooled together and amplified separately for a fraction of the upfront cost of the direct synthesis approach. This strategy opens up the possibility of using different probe sets in any regular research lab.

Targeting more genes with high sensitivity can result in optical overcrowding, which may hinder rolony decoding. Physical expansion of the tissues[37,39,40] has been used as an effective strategy to distance rolonies and reduce overcrowding but it leads to larger imaging areas, longer imaging time and thus lower throughput[37]. A computational solution to the overcrowding problem can vastly increase the throughput. We reasoned that given the size of the rolonies (<1μm)[41] and our pixel size (~0.3μm with 20x objective), each pixel will at most overlap a few rolonies. On the other hand, given that a small fraction of

**Fig. 1 | DART-FISH workflow. a** Schematics of DART-FISH. RNA molecules in a fresh-frozen and formaldehyde-fixed tissue section were reverse-transcribed with primers carrying a 5′ handle with an acrydite modification. A polyacrylamide (PA) gel was cast on the tissue, incorporating the cDNA molecules in the gel matrix. After RNA removal, padlock probes were hybridized to cDNA and circularized, followed by rolling circle amplification (RCA) to create rolonies. Rolonies were further crosslinked to the gel. **b** Imaging DART-FISH samples. Samples went through anchor round imaging followed by decoding rounds. In anchor round imaging, fluorescent probes complementary to the universal sequence and the 5′ cDNA handle, present on all cDNA molecules, were hybridized at room temperature to visualize the distribution of rolonies and the shape of the somas (RiboSoma), respectively. After imaging, the fluorescent probes were stripped and washed away at room temperature. In the subsequent decoding rounds, round-specific decoding probes were hybridized, imaged and stripped. This procedure was repeated $n$ times ($n = 6$ in this example). **c** An example codebook for DART-FISH. Each gene was barcoded such that the corresponding rolonies show fluorescent signal in $k$ ($k = 3$ in this example) rounds of decoding and remain off in other rounds. 5–10% of the codebook consists of empty barcodes that do not have representative padlock probes and were only used for quality control in the decoding pipeline. **d** *SparseDeconvolution* (SpD) decoding algorithm. The intensity of pixels across $n$ rounds of 3-channel imaging was modeled as a weighted combination of the barcodes in the codebook. The decoding was formulated as a regularized linear regression such that most barcodes do not contribute to the observed intensity. **e** Example of decoding by FISH on the PA gel. The lower panel shows the maximum intensity projection of the fluorescent images across 6 decoding rounds and 3 channels (scale bar 5 μm). The upper panel is a cartoon drawing depicting the decoding of a *RORB* spot corresponding to the white square. **f** Lasso maps. Lasso maps are the solutions to the optimization in **d** and represent the gene weights for each of *NRGN, SLC17A7, UCHL1, RORB, TMSB10*, and an Empty barcode in **e** (scale bar 5 μm).

all possible barcodes are used, it may be possible to deconvolve mixtures of barcodes from fluorescent intensity values at the pixel level. To this end we developed the *SparseDeconvolution* (SpD) decoding algorithm: we formalized this deconvolution as a regularized linear regression problem, where barcodes can combine linearly to form the observed pixel intensities and optimized the combinations under a condition that promotes sparsity (Methods, Fig. 1d). We solve this problem for every pixel and obtain initial weight maps for every single barcode (Fig. 1f). This is followed by filtering and aggregating the neighboring pixels to form spots (Supplementary Fig. 2a,b). To control the quality of the deconvolution procedure, we add empty barcodes that are not used in the probe set to the codebook. While the fraction of empty barcodes is 5-8% of used barcodes, the fraction of spots decoded as empty is below 0.25% (empty rate, Supplementary Fig. 2c−e). We compared SpD with existing methods, including a naive algorithm that directly matches pixels to individual barcodes[42] and more sophisticated deconvolution algorithms[42–44]. The results on synthetic data show a complementary performance of SpD to the other deconvolution algorithms while a superior performance to the direct matching algorithm (Supplementary Fig. 2f). The simulations also show that specificity, which is unobserved on real data, is related to empty rate and one can keep specificity high by keeping the empty rate low. With this computational framework, we could mitigate optical overcrowding and increase our throughput by imaging with a 20x objective lens.

## Benchmarking and validation of DART-FISH

To assess the performance of DART-FISH for profiling more than one hundred RNA species in large human tissue sections with fast image acquisition, we applied it to a 10μm-thick, 6.9-by-4.3-mm² fresh-frozen post-mortem human M1C brain section[45]. The anatomy, function, and gene expression of M1C have been widely investigated at the single-cell level[46–50], giving us a well-defined standard to compare across different studies. Note that archived human brain samples represent one of the most challenging sample types for spatial RNA mapping, due to the presence of high autofluorescence[45] and in general, lower RNA quality[51].

We designed 5097 padlock probes to target a selected panel of 121 genes containing known marker genes to resolve the spatial organization of excitatory and inhibitory neurons, as well as non-neuronal cells (Supplementary Data 2). The corresponding codebook followed a 3-on-3-off barcoding scheme. Imaging 6 rounds of decoding, the anchor round and the nuclear stain of this ~30 mm² section of human M1C took about 10 h. After image preprocessing and spot decoding by SpD, we obtained 2,008,260 transcripts (0.2% empty calls with 8 empty barcodes). The expression level of these 121 genes was highly consistent between two replicates (correlation coefficient $r^2 = 0.988$, Fig. 2b), demonstrating a high reproducibility of DART-FISH.

We segmented the cells using RiboSoma, which revealed cell body morphology better than nuclear staining (Supplementary Fig. 4a, b), and assigned the transcripts to the closest cell if the distance to the cell boundary was less than 3μm (Methods, Supplementary Fig. 4c). Other transcripts were discarded from downstream analyses. Among the target genes, we noticed a higher fraction of *MBP* transcripts were found to be outside the cell bodies (93% outside, Supplementary Fig. 4d) while co-localizing with RiboSoma in the extrasomatic space of the cortex (Supplementary Fig. 4e). This observation reflects the local translation of *MBP* transcripts at the axon-glia contact sites[52]. Overall, we detected 26,646 cells with 802,361 transcripts that were assigned to a segmented cell with an average of 30 transcripts and 11 unique genes per cell (Fig. 2c).

To assess spatial specificity of transcript localization, we first inspected the marker genes *SLC17A7* and *SATB2* in excitatory neurons and *GAD1* and *GAD2* in inhibitory neurons. As expected, the *SLC17A7* and *SATB2* transcripts were mainly aggregated in the soma of excitatory neurons with mutual exclusivity to *GAD1* and *GAD2* transcripts in inhibitory neurons (Fig. 2d, e). We then compared the expression of 10 marker genes with the results of RNAscope generated on a parallel M1C tissue section (Methods). As shown in Fig. 2f and Supplementary Fig. 4f, the spatial distribution of these marker genes in the same region demonstrates high concordance between RNAscope and DART-FISH. Specifically, the pan-excitatory neuron marker, *SLC17A7*, showed pronounced enrichment in the L2-L6 cortical layers. *CUX2, RORB*, and *FEZF2* were enriched in supragranular, granular, and infragranular layers of the neocortex, respectively, which is consistent with previous studies[53–57]. The observed localization of *CBLN2* in neocortical layers 2/3 and 5/6 neocortex also agrees with a previous report[58]. Collectively, these results indicate that DART-FISH can specifically map the spatial localization of these marker genes in human M1C.

To estimate the sensitivity of DART-FISH, we selected a similar region of interest (ROI) with equal area between RNAscope and DART-FISH samples and compared the number of transcripts of each gene. We found that the estimated sensitivity ranged from 3.9% to 67.7%, depending on the transcript (Fig. 2g). We correlated our data to the publicly available MERFISH[59] and EEL FISH[60] datasets from the human brain (Pearson's r = 0.755 and 0.750, respectively, Fig. 2h and i), which we consider a high concordance given the differential probing efficiencies between different technologies, and the fact that samples from different regions were used for each technology. In summary, DART-FISH is a reproducible spatial transcriptomic method with the sensitivity and specificity to detect hundreds of RNA species in their spatial context, with the potential for providing biologically meaningful insights to the human brain despite the high natural background autofluorescence.

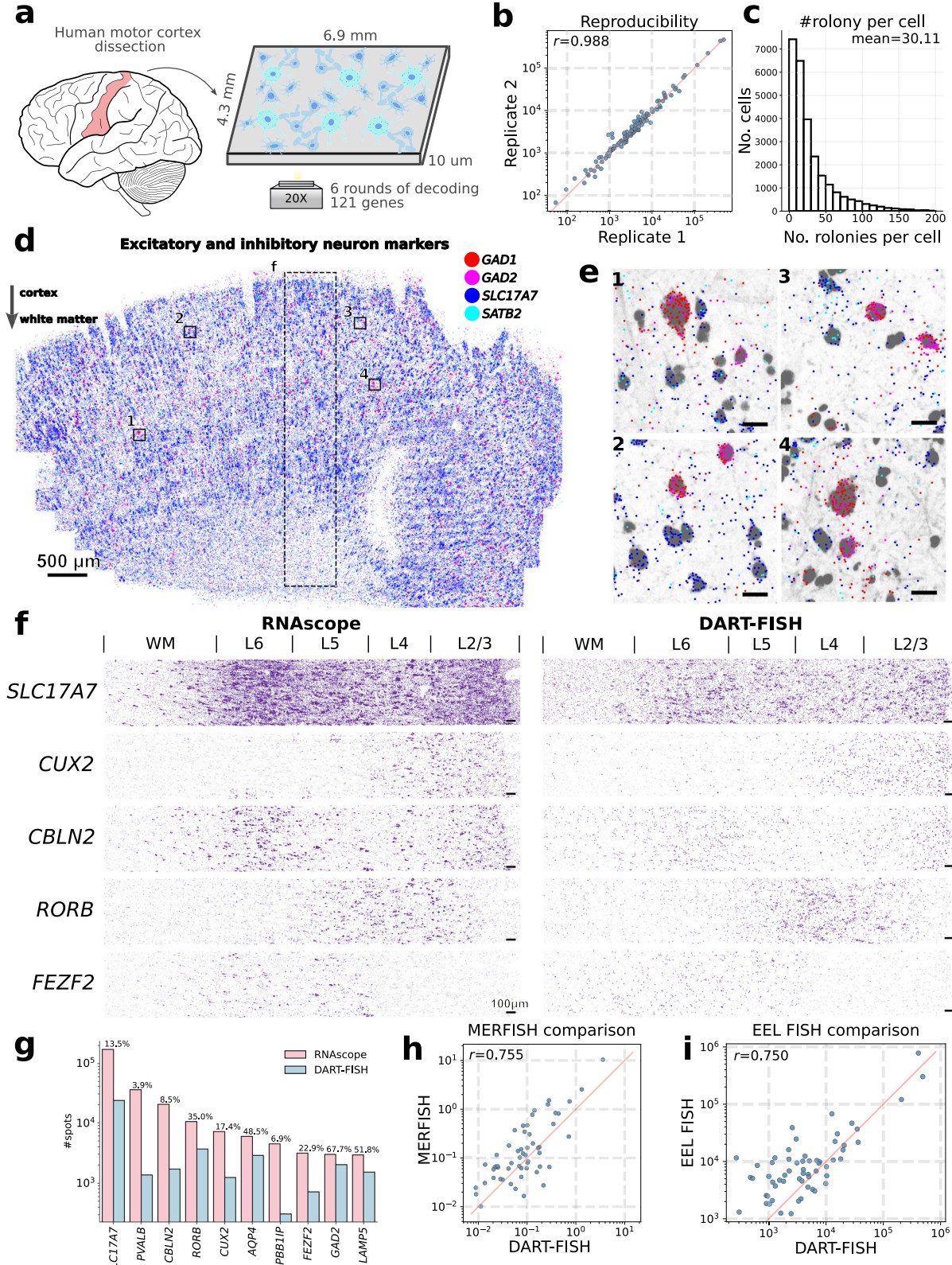

## Organization of cell types in the human primary motor cortex

To assess whether DART-FISH is able to resolve the organization of various cell types of human M1C, we set out to perform cell annotation by performing clustering on DART-FISH cells and matching them to the highest correlated subclass from a recent single-nucleus RNA sequencing (snRNA-seq) reference of M1C[61] (Methods, Fig. 3a and b, Supplementary Fig. 5a, b). We resolved 20 subclasses from the major

excitatory, inhibitory, and non-neuronal cell classes, which constituted 24.3%, 10.6%, and 65.1%, respectively, in the M1C (Fig. 3c–g). For excitatory neuronal subclasses, we successfully detected their laminar distribution, with L2/3 IT neurons localized at the superficial layer of the cortex and L6b/CT neurons deep in the cortex and close to the white matter (Fig. 3b–d), in line with the evolutionarily conserved organization of excitatory neurons in the mammalian M1C[46]. Of note,

**Fig. 2 | Benchmarking DART-FISH on the human M1C. a** Parallel sections were taken from a dissected post-mortem human M1C tissue block. Spatial distribution of 121 genes was measured by DART-FISH with 6 rounds of decoding. **b** Scatter plot showing reproducibility between parallel tissue sections processed independently. Each dot represents the total count of each gene detected in each replicate. **c** The histogram for the number of high quality decoded rolonies per cell. **d** Spatial distribution of excitatory neuron markers (*SLC17A7* and *SATB*) and inhibitory neuron markers (*GAD1* and *GAD2*) in the whole tissue. The dashed rectangular box delineates the ROI in **f**. **e** Zoomed-in views to show the segregation of excitatory and inhibitory markers at single-cell level in 4 ROIs indicated by the black squares in **c**. Scale bars 20 μm. **f** Validation of DART-FISH by RNAscope. Spatial distribution of *SLC17A7, CUX2, CBLN2, RORB* and *FEZF2* across the cortical layers measured by RNAscope (left) and DART-FISH (right). Scale bar 100μm. **g** Quantitative comparison of counts for *SLC17A7, PVALB, CBLN2, RORB, CUX2, AQP4, APBB1IP, FEZF2, GAD2*, and *LAMP5* in DART-FISH and RNAscope in equivalent ROIs. Percentages represent total spots detected in DART-FISH divided by total spots detected in RNAscope multiplied by 100. **h** Comparing DART-FISH and MERFISH[59] (sample H18.06.006.MTG.4000.expand.rep2). Each dot represents the mean count per cell for the 56 shared genes. **i** Comparing DART-FISH and EEL FISH[60] (data from human visual cortex). Each dot represents the total count for one of the 60 shared genes. Source data are provided as a Source Data file.

L6 IT Car3 cells seem to be positioned more superficially than the L6 IT population, consistent with recent observations in human visual cortex and middle temporal gyrus[61,62] (Fig. 3d). In contrast, inhibitory neuronal subtypes generally showed wider spatial gradients along the cortical axis; for instance the Vip population was enriched in layer 2-4 as suggested by previous studies in the mouse[49,63] (Fig. 3b and e). Moreover, we observed some cells belonging to the excitatory neurons and inhibitory neurons localized in the white matter region, which likely are the adult remnants of early generated subplate neurons discovered in previous studies[64,65]. For non-neuronal cells, we observed oligodendrocytes appearing at layer 4 and peaking in the white matter[66] in spite of the uniform distribution of the oligodendrocyte progenitors across the tissue section (OPC, Fig. 3f)[67].

We further assessed whether we could detect short genes (<1.5kb) with DART-FISH. smFISH-based methods rely on tiling sufficiently long RNA molecules with probes to generate detectable fluorescent signals. In contrast, DART-FISH requires only one padlock probe to bind successfully to the target to detect it. To boost our chances for detecting shorter genes, we allowed overlapping targets in our design strategy to obtain more probes for short RNA species[68] (Supplementary Fig. 3b, *NPY* as an example). We compiled a list of 33 differentially expressed genes shorter than 1.5kb comprising well-studied genes as well as less well-known computationally derived marker genes in the brain (Supplementary Data 2). For example, by targeting *SST* (607 nt) and *NPY* (893 nt), we could uncover a rare subclass of inhibitory neurons, Sst Chodl (0.1% abundance, Fig. 3g), specified by the expression of these short neuropeptides (Fig. 3b and h). Sst Chodl cells were found to be enriched in deeper layers, consistent with previous reports[69]. In addition to these short neuropeptides, DART-FISH also detected other short RNA species, including *PCP4* (534nt) and *TMSB10* (461nt) with pronounced localization (Fig. 3h). *PCP4* is reported to be a layer 5-6 marker in the mouse cerebral cortex[70] while *TMSB10* seems to be a deep layer marker gene. To quantify how well the targeted genes performed, we correlated their average expression at the subclass level between DART-FISH and snRNA-seq (Methods, Supplementary Fig. 5c). We found 25 of 33 (75%) of the genes shorter than 1.5kb and 81 of 88 (92%) of the longer genes had higher correlations than 0.5 (Supplementary Data 2). This is similar to a MERFISH data set targeting another region of the human cortex with 250 genes (88% with >0.5 Pearson's correlation, Supplementary Fig. 5c). Taken together, we showed that DART-FISH can accurately map the distribution of all the main neuronal and non-neuronal subclasses in the human brain and can uncover rare cell populations by detecting short genes.

## Mapping cellular neighborhoods in histopathologically abnormal human kidney

To demonstrate the applicability of DART-FISH to a clinically relevant tissue context, we next applied it to the human kidney. The kidney is composed of repetitive functional tissue units, called nephrons, with various closely organized cell types, including endothelial, stromal, immune and epithelial cells that regulate the filtration of the blood as well as other homeostatic functions such as maintaining electrolyte and fluid balance[71] (Fig. 4a, Supplementary Fig. 6a). The homeostatic

interactions between these cell types are perturbed in kidney disease and can lead to fibrosis and decline in kidney function[72]. We recently reported an atlas of cell types in healthy and diseased patients, and identified multiple mal-adaptive cell states that are associated with kidney disease[73,74]. In the same study, we used sequencing-based spatial transcriptomics methods with 10um and 55um resolution to map cellular neighborhoods in healthy and diseased samples, respectively, which lacked the resolution needed to delineate the exact cellular composition, the boundaries and the positioning of cells within the neighborhoods. We reasoned that the high spatial resolution provided by DART-FISH is complementary to the sequencing-based methods and can help define cellular niches more accurately.

Guided by the published single-nucleus reference atlas, we designed a panel of 300 genes with 6299 padlock probes following the 3-on-4-off barcoding scheme, focusing on the major healthy cell types of the kidney, immune cells and cell states implicated in kidney disease (Supplementary Data 3). We then performed DART-FISH on tissue sections from the kidney cortex of a patient with various clinical features, including glomerulosclerosis, interstitial fibrosis, tubular atrophy, and chronic inflammation identified by a pathologist. Our gene panel correctly mapped the spatial organization of cells in different regions of the nephron, including glomeruli and cortical tubules (Fig. 4b). For instance, the transcripts *NPHS2* and *EMCN*, which mark podocytes and glomerular capillary endothelial cells, respectively, are mainly found in the glomerular tuft of the round appearing renal corpuscles. We then compared our data with a Slide-seq dataset from a healthy individual. At the bulk level, the DART-FISH data is correlated with slide-seq (Pearson's r = 0.609) with cells in DART-FISH demonstrating more copies of the targeted genes than Slide-seq beads[73] (median fold-change per gene=2.2 for the top 150 genes in slide-seq, Supplementary Fig. 6b). The comparison also showed upregulation of markers of inflammation in the DART-FISH dataset, consistent with the underlying pathology in our sample (Supplementary Fig. 6b). Hence, the spatial distribution of known kidney marker genes and their overall counts are consistent with kidney biology and prior data.

To find the molecular identity of the cells in the human kidney, cell segmentation was performed using both RiboSoma and nuclear stains. We found RiboSoma to be superior to the nuclear stain in revealing tubular morphology and distinguishing the interstitial cells (Supplementary Fig. 6c). Subsequently, with 30,000 segmented cells with an average of 30 detected transcripts and 20 unique genes per cell (Supplementary Fig. 6d, e, empty rate <0.25% with 15 empty barcodes), the kidney DART-FISH data was annotated to cortical and altered cell types as identified in the single-cell kidney atlas[73] (Fig. 4c, Supplementary Fig. 6f, Supplementary Fig. 7, Methods). These annotated cell types were of the expected relative proportions and showed strong and specific differential expression of corresponding marker genes (Fig. 4d, Supplementary Fig. 6f, Supplementary Fig. 8a). Thus, DART-FISH could confidently resolve >20 cell types and states in the human kidney.

Next, we investigated the neighborhoods formed by the healthy cell types. The complex archetypical structure of the renal corpuscle was successfully recapitulated, with podocytes (POD), glomerular

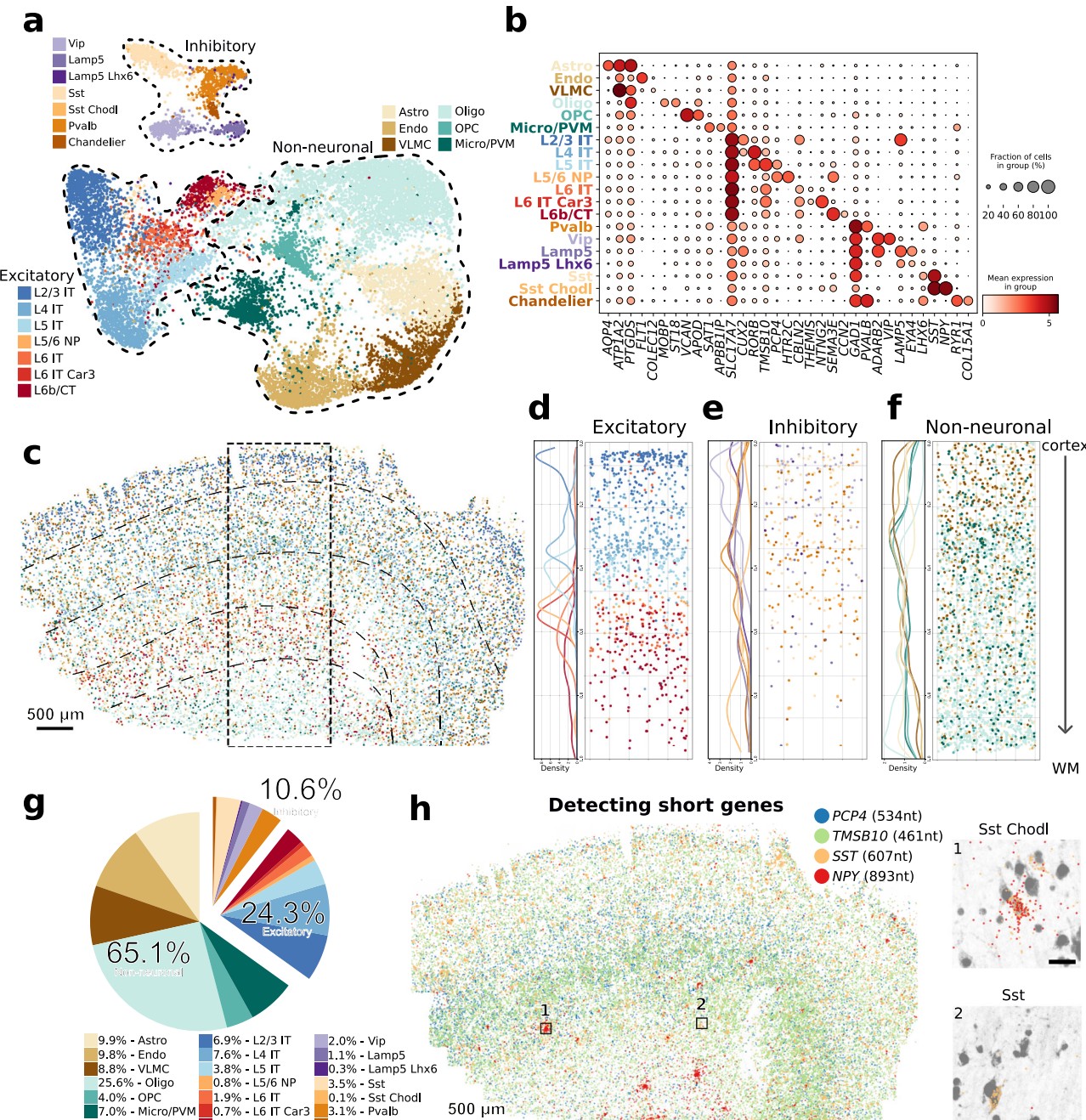

**Fig. 3 | DART-FISH mapping of cell types in the human M1C. a** UMAP plot of all annotated excitatory neurons (L2/3 IT, L4 IT, L5 IT, L5/6 NP, L6 IT, L6 IT Car3, and L6b/CT), inhibitory neurons (Pvalb, Vip, Lamp5, Lamp5 Lhx6, Sst, Sst Chodl, and Chandelier), and non-neuronal (Astro, Endo, VLMC, Oligo, OPC, and Micro/PVM) subclasses. Astro: astrocytes, Endo: endothelial cells, VLMC: vascular and lepto-meningeal cells, Oligo: oligodendrocytes, OPC: oligodendrocyte precursor cells, Micro/PVM: microglia/perivascular macrophages, IT: intratelencephalic, CT: corti-cothalamic, NP: near-projecting. **b** Dot plot of marker gene expression across annotated subclasses. **c** Spatial distribution of all annotated cell types in the entire M1C tissue section from upper cortical layer at the top to the white matter (WM) at

the bottom. The dashed rectangular box delineates the ROI in **d**–**f**. **d**–**f** show the density plot (left) and spatial distribution (right) of excitatory neurons, inhibitory neurons, and non-neuronal subclasses, respectively. **g** Pie chart depicting the relative frequency of annotated subclasses (*n* = 1 section). **h** Spatial distribution of targeted short RNA species *PCP4*, *TMSB10*, *SST*, and *NPY* in the M1C tissue section. *PCP4* and *TMSB10* are layer 5 and layer 5–6 markers, respectively. Sst Chodl cells (0.1% abundance) are *SST⁺ NPY⁺*. Inset 1 shows an example of a Sst Chodl cell, while inset 2 is a *SST⁺ NPY⁻* cell from the more frequent Sst subclass (abundance 3.5%). Inset scale bars 20µm.

capillary endothelial cells (EC-GC) and glomerular mesangial cells (MC) confined within the glomerular tuft, surrounded by parietal epithelial cells (PEC) or the outer layer of the Bowman's capsule and juxtaposed with the renin-secreting cells (REN) in the wall of the arterioles (Fig. 4e, Supplementary Fig. 6a, Supplementary Fig. 7). We also detected medullary rays with the characteristic bundling of the tubules of

cortical thick ascending limb (C-TAL), the S3 segment of proximal tubules (PT-S3) and collecting ducts (Fig. 4f). Further, collecting ducts comprising intermixed principal cells (PC) and alpha- and beta-intercalated cells (C-IC-A and IC-B) could be clearly resolved. These results show that our cell type annotations closely match the known structures within the human kidney.

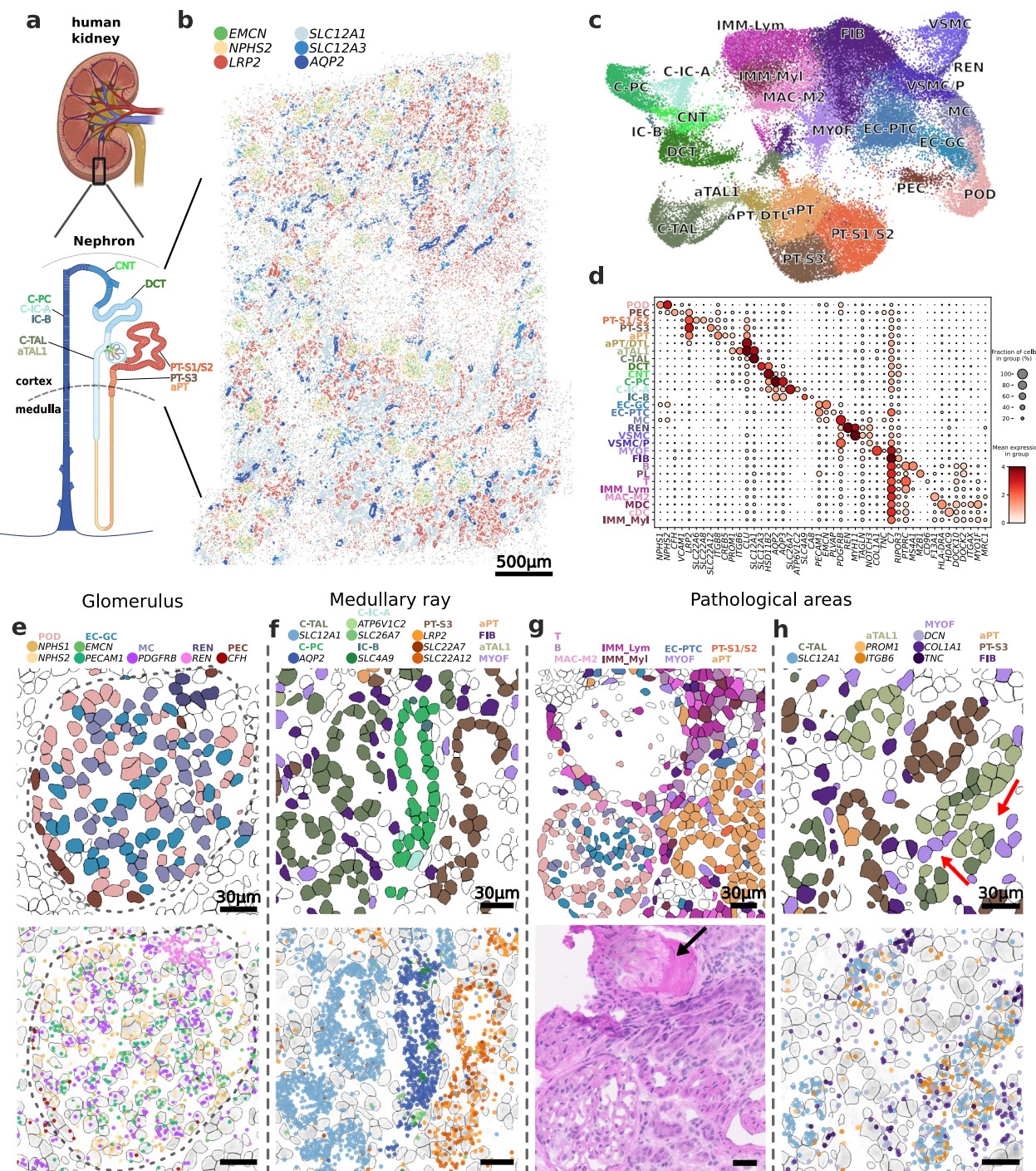

**Fig. 4 | DART-FISH mapping of a diseased human kidney. a** Applying DART-FISH to a 4.9x3.8mm² section from the cortex of the human kidney (adapted from BioRender). The nephron schematics shows the expected epithelial subclasses in the section[101]. **b** The spatial expression of key marker genes for the cortical segments: *EMCN*: glomerular capillary endothelial cells (EC-GC), *NPHS2*: podocytes (POD), *LRP2*: proximal tubules (PT), *SLC12A1*: cortical thick ascending limbs (C-TAL), *SLC12A3*: distal convoluted tubules (DCT), *AQP2*: cortical principal cells of the collecting duct (C-PC). **c** UMAP of all annotated subclasses. PEC: parietal epithelial cells, aPT: altered proximal tubules, DTL: descending thin limbs, aTAL: altered thick ascending limbs, DCT: distal convoluted tubules, CNT: connecting tubules, C-IC-A: cortical intercalated cell type A, IC-B: intercalated cell type B, EC-PTC: peritubular capillary endothelial cell, MC: mesangial cell, REN: renin-positive juxtaglomerular granular cell, VSMC: vascular smooth muscle cell, VSMC/P: vascular smooth muscle cell/pericyte, FIB: fibroblast, MYOF: Myofibroblast, MAC-M2: M2 macrophage,

IMM-Lym: lymphoid cell, IMM-Myl: myeloid cell. **d** Dot plot of marker gene expression for the annotated subclasses. **e** An example of a glomerulus with part of the juxtaglomerular apparatus. (top) cells colored by the annotated subclass, (bottom) marker genes corresponding to the subclasses. Each dot represents one rolony. Dashed line delineates the boundary of the renal corpuscle. **f** Example of a medullary ray with a bundle of TALs, PT-S3, and collecting ducts. Note that for clarity, some cell types, i.e., aPT, FIB, aTAL1 and MYOF are plotted (top) but their corresponding marker genes are omitted (bottom). **g** Example of a pathological niche with inflammation, a sclerosed glomerulus and altered proximal tubule cells adjacent to a more normal glomerulus (top). The same area on an H&E-stained parallel section from the same tissue block confirms the decellularization and inflammation observed in DART-FISH. The black arrow points to the sclerotic glomerulus. **h** Example of a pathological niche composed of aTAL1 cells and myofibroblasts. Red arrows point toward densities of MYOF and aTAL1 cells.

To compare the tissue morphology obtained from DART-FISH with a clinically relevant histological stain, we performed Hematoxylin and Eosin (H&E) staining on a parallel section from the same tissue block. In an area with putative inflammation on the H&E slide, we observed an abundance of immune cells of both lymphoid and myeloid origin on the DART-FISH section (Fig. 4g). These immune cells surround a sclerotic glomerulus, which in contrast to a more normal glomerulus, is depleted from cells and is instead fibrotic (shown by an arrow in Fig. 4g). In DART-FISH, this phenomenon can be clearly detected by contrasting the low cell numbers revealed by RiboSoma and the physically occupied space through the accompanying transmitted light image (Supplementary Fig. 6h). Thus, by paired H&E staining we showed that DART-FISH can capture different pathological phenomena with a molecular resolution beyond that of the traditional histology.

In addition to healthy cell types, DART-FISH was also able to reveal distinct pathological cell states. This includes a population of myofibroblasts (MYOF) expressing matrisome genes, including *COL1A1*, *TNC*, *DCN* and *POSTN*, suggestive of their ECM-producing role in kidney fibrosis (Supplementary Fig. 8b)[73,75]. Furthermore, we detected altered PT (aPT) and TAL (aTAL1) populations, both of which expressed *PROM1*, in line with recent findings[73,76]. To determine whether these pathological cell states form distinctive niches, computational methods were applied to find pairs of cell types that show enrichment in their spatial colocalization[77]. Interestingly, in neighborhoods around MYOFs, there was an increased presence of aTAL1 cells compared to C-TAL and aPT (Fig. 4h, Supplementary Fig. 6i). This observation indicates a possible interplay between the maladaptive repair of TALs and fibrosis. We speculate that there are a variety of cellular neighborhoods associated with adaptive repair and fibrosis that could be defined through further studies. All in all, these results demonstrate how DART-FISH as a single-cell resolution spatial transcriptomic technique can be used to interrogate neighborhoods of cell types and states defined by single-cell RNA sequencing studies in diseased human tissues.

## Discussion

In this study we introduced DART-FISH, a high throughput RNA in situ mapping technique, and demonstrated its application to human tissues, even with high native autofluorescence background. In the human brain, we recovered the spatial distribution of 20 cell types from the 3 main cell classes. This included the laminar organization of the excitatory neurons in the cortex and the broader layer-specificity of inhibitory neurons, and the ubiquity of the non-neuronal cells across the brain cortex. We also profiled a sample from a histopathologically abnormal human kidney and demonstrated identification of rare cells such as *REN*-producing cells, the intricate functional niches, and quantified the interactions between pathological cell states.

DART-FISH is a cost-effective technology capable of fast decoding on relatively large tissue sections. Using our protocol for padlock probe production from oligo pools, the cost of synthesis per gene scales sublinearly with the number of genes. Hence, oligo pricing will not hinder scaling the probe set to tens of thousands of transcripts. Moreover, DART-FISH does not need any specialized equipment for neither rolony generation nor decoding. The decoding process is relatively fast because it depends on the diffusion and hybridization of very short oligos and a strong signal can be obtained by 5-10 min of incubation with the fluorescent decoding probes at room temperature. Likewise, stripping and washing away the unbound decoding probes is straightforward and fast at room temperature. This process can be performed on a stationary glass-bottom petri dish or a coverslip mounted on a microscope and does not require reaction chambers or flowcells with sophisticated temperature control. The large size and the bright signal of the rolonies permit the use of 20x objective lenses

for decoding, which makes it possible to image centimeter-sized samples in a manageable time with an ordinary confocal microscope.

What distinguishes DART-FISH from other techniques of a similar class is how the cDNA molecules are treated[35,36]. We demonstrated here that embedding the cDNA molecules in a polyacrylamide gel significantly enhances the retention of the cDNA throughout the rolony generation procedure and increases the sensitivity, a point not taken into account in previously published methods. Additionally, we introduced RiboSoma, a cDNA labeling technique, as a cell morphology marker which reveals more information about cell bodies than nuclear stains. We anticipate that this tool can be highly useful for cell body segmentation, particularly in thicker samples.

RCA-based in situ detection systems are prone to optical and physical overcrowding as more and more genes are detected with higher efficiency. To mitigate this issue, we developed a computational method (SpD) that used the redundancy in the barcode space to deconvolve mixed barcodes from single pixels. This strategy improved our decoding efficiency compared to naive decoding methods[42]. The utility of this method increases with higher redundancy in the barcode space by creating longer barcodes with more "on" cycles, and careful assignment of barcodes to genes such that genes that tend to co-express in the same cell types have unique barcode combinations. In addition, more sophisticated deconvolution methods that share information between neighboring pixels can potentially improve decoding efficiency[43,44,78]. As the field is moving towards detecting more genes in parallel, pixel-based deconvolution methods like SpD could become increasingly relevant.

Although we have only tested DART-FISH on fresh-frozen tissue sections, we think it should be compatible with other tissue preservation methods as long as the RNA integrity is well-preserved. We have found tissue quality to be a critical source of variability across experiments and hence should be controlled by meticulous preparation and handling of tissue blocks. Future studies that systematically evaluate various preservation methods for post-mortem human tissues will be key to advancing the field. Note that different fixation methods, as well as different tissue types, may require optimization of the tissue processing steps (e.g., permeabilization) before reverse-transcription. RiboSoma can be a helpful guide through this optimization, as the overall intensity of the signal and the morphological patterns can be used to compare different treatment conditions.

Due to its streamlined nature and simplicity, the basic DART-FISH chassis described here can be effectively extended in multiple ways. The workflow can be combined with antibody staining, for instance, to target extracellular factors such as matrix proteins and cell-cell communication molecules to enhance the definition of cell-cell interactions in pathological niches[79]. The thickness of tissue sections could be increased for higher resolution mapping of neighborhoods and cell connectivities; while increasing section thickness to 20-30μm should be readily achievable, other strategies in sample mounting and handling may be necessary to increase the diffusion into even thicker sections (>100μm)[80]. Padlock probes could also be designed to anneal directly to mRNA followed by circularization using an RNA-mediated DNA ligase, which would skip the cDNA synthesis and can improve the detection sensitivity.

## Methods
### Human tissue samples

**Human brain.** Human Brain tissue was obtained from the University of Washington Biorepository and Integrated Neuropathology (BRaIN) Laboratory under UW School of Medicine and HIPAA compliance. Informed consent was obtained for the use of data and samples. One donor brain with postmortem interval ≤12 h and RIN score ≥7 was selected for DART-FISH assay. Regions were identified and isolated utilizing architectural landmarks, aided by the Allen Brain Human Brain Atlas[81]. Multiple parallel 10-μm-thick cryosections were taken from the

tissue block and mounted onto vectabond-coated 24 x 60 mm No.1.5 coverslips (Azer Scientific, 1152460). Brain cryosections were stored at −80 °C until use.

**Human kidney.** Kidney tissue was obtained from the Kidney Translational Research Center (KTRC) biorepository under a protocol approved by the Washington University Institutional Review Board (IRB 201102312). Informed consent was obtained for the use of data and samples. The kidney tissue was dissected from the whole kidney and freshly frozen in Optimal Cutting Temperature embedding media in cryomolds on a liquid nitrogen chilled metal block and stored at −80 °C until ready for experimental use[74]. 10-μm-thick sections were cut from the frozen blocks for DART-FISH and flanking sections were used for histopathological assessment by a renal pathologist.

## Reagents and enzymes
All reagents were listed as in Supplementary Data 1.

## Gene selection
A list of genes was selected based on differential expression analysis of snRNA-seq data from human primary motor cortex[46,48,50] and a few curated marker genes were added manually to target 121 genes in the human M1C. Human kidney gene selection was performed by gpsFISH[82,83] to distinguish subclass level 2 annotation in our kidney reference atlas[73]. snRNA-seq data from the kidney reference atlas with cell type annotation at subclass level 2 was used as input of gpsFISH. Curated marker genes from prior knowledge were also included as input. The size of the gene panel was set to 300. We ran the optimization for 100 iterations to ensure convergence although the optimization converged around iteration 50.

## Probe design and production
**DART-FISH probe design.** For short genes (length < 1.5kb), we defined the constitutive exon as the union of all isoforms in GencodeV41. For other genes, the constitutive exons were defined as regions in RefSeq where at least (33% for the brain, 50% for the kidney) of isoforms overlap. We used a modified version of ppDesigner[38] (https://github.com/Kiiaan/sppDesigner) to find padlock target sequences along the constitutive exons. ppDesigner was run on two settings: 1) no overlap between probes allowed, 2) overlap of up to 20nt allowed. Individual arms were constrained between 17nt and 22nt long with the total target sequences no longer than 40nt. The resulting target sequences were aligned to GRCh38/hg38 with BWA-MEM[84] and sequences with MAPQ < 40 or secondary alignment were removed. We further removed probes that have GATC (DpnII recognition site). For the brain, a maximum of 50 probes per gene were selected prioritizing the non-overlapping set. For the kidney, a maximum of 40 probes per gene were selected with no overlap. Finally, the target sequences were concatenated with amplification primer sequences, universal sequence, and gene-specific decoder sequences to produce final padlock probe sequences (Supplementary Fig. 3c) and were ordered as an oligo pool from Twist Bioscience (South San Francisco, CA). Amplification primer pairs pAP1V41U and AP2V4 were used for the kidney probe set, while the brain probe set was amplified with AP1V7U and AP2V7 primer pair (Supplementary Data 1).

To select a set of barcodes, we computationally created all possible barcodes in the compact format: an $n$ digit barcode with "1", "2" and "3" representing each of the three fluorescent channels and "0" indicating off cycles. For example, the barcode for *RORB* in Fig. 1c is "132000" in the 6-digit format. This amounted to 480 and 840 multi-color barcodes for brain and kidney, respectively. We then used a brute force algorithm to find the largest subset of barcodes, $Q$, in which every pair had a Hamming distance > 2. Followed by this, we created a graph, $G$, in which every possible barcode is a node, and pairs of nodes are connected with edges if their Hamming distance is 1. We then

found a maximal independent set (MIS, networkx v2.6.2) that included the nodes in $Q$. This method ensures that every pair of barcodes in the MIS have Hamming distance >1. Because the algorithm for finding MIS is random, we ran it 20,000 times and selected the largest MIS across the runs. For the brain, the MIS consisted of 159 barcodes, 121 of which were randomly assigned to the genes. For the kidney, the MIS had 269 barcodes. We randomly added 31 additional barcodes and counted the number of edges of the induced subgraph of $G$ with the selected nodes. We repeated this selection 20,000 times and proceeded with the run with the lowest edge count. 300 genes were randomly assigned to these barcodes.

**Large-scale padlock probe production.** A step-by-step protocol can be found on protocols.io (dx.doi.org/10.17504/protocols.io.n92ldm3 pxl5b/v1) and is illustrated in Supplementary Fig. 3c. Briefly, oligo pools were PCR amplified on a 96-well plate (10pM per reaction) using KAPA SYBR fast and 0.4μM of each amplification primer (pAP1V41U and AP2V4 for kidney, AP1V7U and AP2V7 for brain, Supplementary Data 1, Supplementary Fig. 3c) until plateau. The PCR products were pooled and concentrated with ethanol precipitation and further purified using QIAquick PCR purification kit (Qiagen 28106).

For the brain probe set, the purified amplicons were divided into parallel reactions (about 5ug each) and were digested with Lambda Exonuclease (0.5U/ul) in 1x buffer (NEB M0262L) at 37 °C for 2 h and purified using Zymo ssDNA/RNA clean & concentrator kit following manufacturer's instructions (Zymo D7011). Next, the single-stranded probes were further digested with 5 units of USER enzyme (NEB M5505L) in 1x DpnII buffer at 37 °C for 3 h. Subsequently, for each reaction we added DpnII guide oligo (Supplementary Data 1) to final concentration of 5uM in 1x DpnII buffer, heated the mix to 94 °C for 2 min, cooled to 37 °C and added 50 units of DpnII in 1x DpnII buffer and incubated for 5 h. Finally, probes were size-selected using a TBE-Urea gel.

For the kidney probe set, DpnII digestion was performed after PCR. In detail, the purified amplicons were divided into parallel reactions (about 5ug each) and were digested with DpnII (1U/ul) in 1x NEBuffer DpnII (NEB R0543L) at 37 °C for 3 h and purified with QIA-quick PCR purification kit. The purified products were digested with Lambda Exonuclease (0.5U/ul) in 1x buffer (NEB M0262L) for 2 h and purified with Zymo ssDNA/RNA clean & concentrator kit. Finally, the library was digested with USER (0.0625U/ul, M5505L) in 1x NEBuffer DpnII in parallel reactions (about 2.5ug each) for 6 h at 37 °C followed by 3 h at room temperature and purified with Zymo ssDNA/RNA clean & concentrator kit.

## DART-FISH
The overall workflow, including reverse transcription, cDNA cross-linking, padlock probe capture, RCA, rolony crosslinking and image acquisition, is illustrated in Fig. 1. A step-by-step protocol can be found at protocols.io (dx.doi.org/10.17504/protocols.io.e6nvwjxnzlmk/v1).

**Reverse transcription and cDNA crosslinking.** Tissue sections were fixed in 4% PFA in 1x PBS at 4 °C for 1 h, followed by two 3-minute washes with PBST (1x PBS and 0.1% Tween-20). Then, a series of 50%, 70%, 100%, and 100% ethanol were used to dehydrate the tissue sections at room temperature for 5 min each. Next, tissues were air dried for 5 min and in the meantime silicone isolators (Grace Bio-Labs, 664304) were attached around the tissue sections. Then, the tissue sections were permeabilized with 0.25% Triton X-100 in PBSR (1x PBS, 0.05U/μl Superase In, 0.2U/μl Enzymatics RNase Inhibitor) at room temperature for 10 min, followed by two chilled PBSTR (1x PBS, 0.1% Tween-20, 0.05U/μl Superase In, 0.2U/μl Enzymatics RNase Inhibitor) washes and a water wash. Next, the sections were digested with 0.01% pepsin in 0.1 N HCl (pre-warmed 37 °C for 5 min) at 37 °C for 90 s and washed with chilled PBSTR twice. Afterwards, acrydite-modified dT

and N9 primers (Acr_dc7-AF488_dT20 and Acr_dc10-Cy5_N9, Supplementary Data 1) were mixed to a final concentration of 2.5 μM with the reverse-transcription mix (10U/μL SuperScript IV (SSIV) reverse transcriptase, 1x SSIV buffer, 250 μM dNTP, 40 μM aminoallyl-dUTP, 5 mM DTT, 0.05U/ul Superase In and 1U/μL Enzymatics RNase inhibitor). The sections with the mix were incubated at 4 °C for 10 min and then transferred to a humidified 37 °C oven for overnight incubation. After reverse transcription, tissue sections were washed with chilled PBSTR twice and incubated in 0.2 mg/mL Acryloyl-X, SE in 1x PBS at room temperature for 30 min. Then, the tissue sections were washed once with PBSTR, followed by incubation with 4% acrylamide solution (4% acrylamide/bis 37:1, 0.05U/μL Superase-In, and 0.2U/μL RNase inhibitor) at room temperature for 30 min. Subsequently, the acrylamide solution was aspirated and gel polymerization solution (0.16% Ammonium persulfate and 0.2% TEMED in the 4% acrylamide solution) was added. Immediately, the tissues were covered with Gel Slick (Lonza #50640)-treated circular coverslips of 18 mm diameter (Ted Pella, 260369), transferred to an argon-filled chamber at room temperature and incubated for 30 min. After gel formation, the tissue sections were washed with 1x PBS twice and the coverslip was gently removed with a needle. At this point, the cDNA is crosslinked to the polyacrylamide gel.

**Padlock probe capture.** After cDNA crosslinking in gel, remaining RNA was digested with RNase mix (0.25U/μL RNase H, 2.5% Invitrogen RNase cocktail mix, 1x RNase H buffer) at 37 °C for 1 h followed by two PBST washes, 3 min each. The padlock probe library was mixed with Ampligase buffer. Then, the mixture was heated to 85 °C for 3 min and cooled on ice. Subsequently, the mixture was supplemented with 33.3U/μL Ampligase enzyme such that the final concentration of padlock probe library was 180 nM and 100 nM for the kidney and brain probe set, respectively, in 1x Ampligase buffer. Finally, the samples were incubated with probes at 37 °C for 30 min, and then moved to a 55 °C humidified oven for overnight incubation.

**RCA and rolony crosslinking.** After padlock probe capture, the tissue sections were washed with 1x PBS three times, 3 min each and hybridized with RCA primer solution (0.5 μM rca_primer, 2x SSC, and 30% formamide) at 37 °C for 1 h. Then, the tissue sections were washed with 2x SSC twice and incubated with Phi29 polymerase solution (0.2 U/μL Phi29 polymerase, 1x Phi29 polymerase buffer, 0.02 mM aminoallyl-dUTP, 1 mg/mL BSA, and 0.25 mM dNTP) at 30 °C in a humidified chamber for 7 h. Afterwards, the tissue sections were washed with 1x PBS twice, 3 min each and the rolonies were crosslinked with 5 mM BS(PEG)9 in 1x PBS at room temperature for 1 h. The crosslinking reaction was terminated with 1M Tris, pH 8.0 solution at room temperature for 30 min. Finally, samples were washed with 1x PBS twice and stored in a 4 °C fridge until image acquisition.

**Image acquisition**

**Human Brain.** Human brain tissue sample was stained with 1x True-Black in 70% ethanol at room temperature for 2 min to reduce the lipofuscin autofluorescence and washed with 1x PBS three times for 3 min each before imaging. For the anchor round imaging, a mixture of anchor round probes, including DARTFISH_anchor_Cy3, dcProbe10_ATTO647N, and dcProbe7_AF488 probes, were diluted to 500nM in 2x SSC and 30% formamide. Then, the samples were stained with anchor round probes at room temperature for 5 min and washed with 1 mL washing buffer (2x SSC, 10% formamide and 0.1% Tween-20) twice for 2 min each prior to imaging. The samples were immersed in 1 mL imaging buffer (2x SSC and 10% formamide) during imaging. For decoding imaging, each imaging cycle started with incubating samples with stripping buffer (2x SSC, 80% formamide, and 0.1% Tween-20) at room temperature for 5 min, washed with washing buffer twice for 2 min each, stained with a mixture of AlexaFluor488, Cy3, and

ATTO647 fluorophore-labeled decoding probes (dcProbe0-AF488, dcProbe0-Cy3, and dcProbe0-ATTO647N as an example for round 1) in 2x SSC and 30% formamide for 10 min, and washed with washing buffer three times for 2 min each. Then, the samples were immersed in 1 mL of imaging buffer while imaging. After the last cycle of decoding imaging, DRAQ5 staining (5 μM, room temperature, 10 min) was performed for nuclei segmentation. Z-stack images were acquired by a resonant-scanning Leica TCS SP8 confocal microscope with 20x oil-immersion objective (NA = 0.75), pinhole size of 1 airy unit, pixel size of 284 nm x 284 nm (zoom=2) with 1024 x 1024 pixels per image, and 2 line averaging with 26 z-stacks (step size 1μm).

**Human Kidney.** The same fluorescent probes were used as in the human brain imaging in this order: anchor round, decoding rounds 1 to 7, DRAQ5 nuclear staining. All hybridizations were performed with 500nM of each of the fluorescent oligos in 2x SSC and 30% formamide for 15 min. Following hybridization, the unbound probes were washed with 4–5 washes with PBST each 2–3 min. Imaging was performed in PBST on a resonant-scanning Leica SP8 with a 20x oil-immersion objective (NA = 0.75), pinhole size of 2 airy units, pixel size of 366 nm x 366 nm (zoom=1.55) with 1024 x 1024 pixels per image, 3 line averaging, with 24 z-stacks (step size 2.5um). After each imaging round, stripping was performed with 80% formamide in 2x SSC and 0.1% Tween-20, 3 times each 3-5 min, followed by 2 quick washes with PBST to prepare for the next hybridization.

**RNAscope**

**Sample preparation.** RNAscope HiPlex 50x probe stocks of human *SLC17A7, RELN, CUX2, RORB, CBLN2, FEZF2, GAD2, PVALB, LAMP5, PLP1, AQP4,*and *APBB1IP* with HiPlex12 Reagent Kit v2 (488, 550, 650) Assay (ACD, 324419) were purchased from Advanced Cell Diagnostics (ACD). The 50x probe stocks and RNAscope HiPlex diluent were warmed at 40 °C for 10 min. The pre-warmed 50x probe stocks were pooled and diluted to 1x with pre-warmed RNAscope HiPlex diluent before use. RNAscope experiments were carried out according to the manufacturer's protocol (document number UM324419) with slight modifications for post-mortem human brain tissue. Briefly, the human brain tissue sections were fixed with 4% PFA in 1x PBS at 4 °C for 1 h and dehydrated with a series of 50%, 70%, 100%, and 100% ethanol at room temperature for 5 min each. Then, silicone isolators of 20 mm in diameter (Grace Bio-Labs, 664304) were applied around the tissue sections and the tissue sections were slightly digested with 5 drops of Protease IV at room temperature for 30 min and washed with 1x PBS for 2 min twice. Subsequently, enough volume of 1x pooled probes was added to cover the tissue sections entirely and the probe hybridization was performed in the 40 °C HybEZ Hybridization System for 2 h. Then, the tissue sections were washed with 1 mL 1x wash buffer at room temperature for 2 min twice. Later, the tissue sections were hybridized with RNAscope HiPlex Amp1, incubated in the 40 °C HybEZ Hybridization System for 30 min, and washed with 1x wash buffer at room temperature for 2 min twice. Afterwards, we followed the same process to hybridize the tissue sections with RNAscope HiPlex Amp2 and RNAscope Hiplex Amp3. Finally, we incubated the tissue section with freshly prepared 5% HiPlex FFPE reagent at room temperature for 30 min and washed the tissue sections with 1 mL 1x wash buffer at room temperature for 2 min twice prior to image acquisition.

**Image acquisition.** The tissue sections with silicone isolators were mounted on the stage of a Leica SP8 confocal microscope and 4 cycles of imaging were performed to image 12 RNA species. In the first imaging cycle, RNAscope HiPlex Fluoro T1-T3 probes were prewarmed at 40 °C, added to cover the tissue sections entirely, and hybridized with the tissue sections for 5 min thrice. After probe hybridization, the tissue sections were washed with 1 mL 1x wash buffer at room temperature for 2 min twice and immersed in 1 mL 4x SSC buffer. Z-stack

images were acquired by Leica TCS SP8 confocal microscope with 63x oil-immersed objective (NA 1.4) and pixel size of 113 nm x 113 nm. Then, the fluorophores were cleaved with freshly prepared 10% cleaving solution (100 µL cleaving solution diluted with 900 µL 4x SSC buffer) at room temperature for 15 min and the tissue sections were washed with 0.5% PBST (1x PBS with 0.5% Tween-20) at room temperature for 2 min twice. The fluorophore cleaving process was repeated once to ensure the fluorophores were removed entirely. This process was repeated 3 more rounds to image RNAscope HiPlex Fluoro T4-T12. An additional "Empty" cycle was performed to image the auto-fluorescence of the human brain tissue without any probes. After the last imaging cycle, we added 80% formamide in 2x SSC buffer to remove RNAscope probes completely and stained the nuclei with 5 µM DRAQ5 at room temperature for 10 min.

**RNAscope data processing.** RNAscope data was processed with the DART-FISH pipeline with one modification. The images from the "Empty" cycle were subtracted from all RNAscope images to remove the autofluorescence.

### DART-FISH data processing (DF3D)
The DART-FISH datasets were processed by our custom pipeline. The source codes of the pipeline can be found in this Github page (https://github.com/Kiiaan/DF3D). Raw z-stack images with 4 channels (3 fluorescent channels and brightfield) from the microscope were registered to a reference round by affine transformation implemented in SimpleElastix[85] using the brightfield channel as the anchor. Then, each field of view (FOV) underwent decoding to obtain a list of candidate spots. Spots from all FOVs were pooled and filtered (See Sparse deconvolution (SpD) decoder for more details). To obtain the global position of the rolonies, the FOVs were stitched by applying FIJI's[86] Grid/Collection Stitching plugin[87] (in headless mode) to the registered and maximum-projected brightfield images. Note that the theoretical positions of the FOVs, defined by the microscope, were used as initial positions for stitching.

Cell boundaries were segmented with Cellpose (v2.1.1)[88,89]. The "cyto" model in Cellpose was fine tuned on each tissue by manually segmenting a handful of composite images of DRAQ5 (nuclei channel) and N9 cDNA stain (cyto channel) using the package's graphical user interface.

**Sparse deconvolution (SpD) decoding.** In DART-FISH, each gene is represented by a barcode that can be read out in $n$ rounds of 3-channel imaging. Each barcode is designed to emit fluorescence (be "on") in exactly $k$ rounds, each time in a single fluorescent channel and stay "off" in other rounds. We concatenate the rounds and channels and represent the barcodes as $3n$-dimensional vectors. In other words, barcode $i$ is represented by vector $x_i$ in which 1's are placed where "on" signal is expected, and 0's everywhere else. The codebook matrix $X$ ($3n$x$N$) is then defined as $X = [\mathbf{x}_1, \mathbf{x}_2, ..., \mathbf{x}_N]$, where $N$ is the total number of barcodes. In the same way, for every pixel we concatenate the fluorescent intensity values (scaled between 0 and 1) to create a $3n$-dimensional vector $\mathbf{y}$.

The fluorescence signal at each pixel can be sourced from more than one rolony if the distance between neighboring rolonies is smaller than the optical resolution of the imaging system, or if 3-dimensional stacks are analyzed as maximum-projected 2D images. Nevertheless, because of physical constraints, only a handful of rolonies are expected to be the source of signal to each pixel. In this regard, because of the redundancy in the barcode space, combinations of barcodes in one pixel can be decomposed into their original composing barcodes. We formulated this problem as a regularized linear regression problem where a weighted sum of a few barcodes creates the observed signal intensity, where the vector $\mathbf{w} = [w_1, w_2, ..., w_N]^T$ shows the contribution of each barcode (Fig. 1d) with most $w_s (1 \leq s \leq N)$ elements equal to 0.

We initially used lasso to solve this problem ($\alpha' = 0$ in Fig. 1d) to promote the sparsity of $\mathbf{w}$, but later decided to use elastic net with a non-zero value for $\alpha'$ that is much smaller than $\alpha$ ($\alpha' = \alpha/100$) to increase stability. We call the solution to this problem $\hat{\mathbf{w}}_{lasso}$. Note that, we constrain the problem to positive weight values ($\hat{\mathbf{w}}_{lasso_s} \geq 0$ for every $s$). The regression problems are solved for all the foreground pixels ($||\mathbf{y}||_2 > 0.25$) individually. For every barcode $i$, we can construct an image with the estimated weight values as pixels: 0 for background and rejected pixels, and non-zero values from $\hat{\mathbf{w}}$. We call these images weight maps. Figure 1f shows weight maps constructed with $\hat{\mathbf{w}}_{lasso}$ which have not been filtered.

With our current barcode space, we can only confidently decompose bi-combinations. Hence, for every instance of the elastic net problem, we applied an elbow filter and accepted the solution only when the top one or two weights were significantly larger than other weights.

In more detail, for every pixel, the weights in $\hat{\mathbf{w}}_{lasso}$ are sorted in decreasing order. If the second largest weight is smaller than half of the top weight, then the top weight passes the elbow filter. Otherwise, if the third largest weight is smaller than 30% of the largest weight, the top two weights pass the elbow filter. All the values that do not pass the filter are set to zero. For accepted solutions, we performed an ordinary least square (OLS) regression using the top one or two weights to obtain unbiased weights ($\hat{\mathbf{w}}_{OLS}$). Supplementary Fig. 2a shows weight maps constructed with $\hat{\mathbf{w}}_{OLS}$ (OLS maps) after applying a Gaussian smoothing.

**Estimating channel-specific coefficients.** So far, we have assumed that pixel intensities from different rounds and fluorescent channels all have the same scale and distribution. However, there is usually a variation among rounds and fluorescent channels, with some channel-rounds being brighter than others. To account for this effect, we model the channel-specific variations as a multiplicative factor that connects the weights at each pixel to intensities: $\mathbf{y} = \mathbf{c} \odot X\mathbf{w}$ where $\mathbf{c} = [c_1, c_2, ..., c_{3n}]^T$ is the channel coefficient vector and $\odot$ denotes element-wise multiplication. Suppose for a set of pixels $\mathbf{y}^{(1)}, \mathbf{y}^{(2)}, ..., \mathbf{y}^{(P)}$ the true barcode weights $\mathbf{w}^{(1)}, \mathbf{w}^{(2)}, ..., \mathbf{w}^{(P)}$ are given. For pixel $i$ and channel $j$, we could write: $y_j^{(i)} = c_j \sum_{b=1}^{N} X_{jb} w_b^{(i)} = c_j \sum_{b=1}^{N} (\mathbf{x}_j)_b w_b^{(i)}$ where $(\mathbf{x}_j)_b$ shows the $b$'s element of the $j$'s barcode. In this case, each $c_j$ can be estimated by solving an OLS problem between $y_j^{(\cdot)}$ and $\sum_{b=1}^{N} (\mathbf{x}_j)_b w_b^{(\cdot)}$. Conversely, if the channel coefficients are given, we can set up the decoding problem with normalized intensities: $\bar{\mathbf{y}} = \mathbf{y}/\mathbf{c} = X\mathbf{w}$ with $/$ being element-wise division. We estimate the channel coefficients in an iterative manner following the algorithm below:

1. Initialize $\mathbf{c} = \mathbf{1}$ (no channel variation)
2. Take a random sample of foreground pixels
3. Normalize the pixel intensities in the sample with $\mathbf{c}$
4. Run SpD on the normalized pixels
5. Keep pixels with one dominant unsaturated weight (weight in range 0.1 and 0.5) and obtain unbiased weights through OLS
6. Update the values of $\mathbf{c}$ by solving $3n$ OLS problems
7. Repeat steps 3–6 $n_{iter}$ times

We do this procedure for 2 iterations and apply the obtained values when decoding all fields of view.

**Setting the elastic net regularization parameter.** Because of physical constraints, the solution to the deconvolution problem must be sparse, i.e., only a few non-zero weights should explain the observed intensities. The sparsity of the solution is directly controlled by the L1 regularization term, $\alpha$ (Fig. 1d). For a given pixel $\mathbf{y}$, higher values of $\alpha$ shrink the estimated weights ($||\hat{\mathbf{w}}_{lasso}||_1 \rightarrow 0$). Conversely, lower values of $\alpha$ allow more weights to be non-zero and $||\hat{\mathbf{w}}_{lasso}||_1$ to grow larger. In fact, one can show if the L2 regularization term, $\alpha' = 0$, the largest weight to be undetected for a pixel made purely from one barcode is

$w_{max} = \frac{3n}{k}\alpha$ [90]. For instance, given $\alpha = 0.05$ and codebook parameters $n = 6$, $k = 3$, then $w_{max} = 0.3$. This means that a pixel composed of one barcode needs to have an underlying intensity >0.3 to get a non-zero $\widehat{\mathbf{w}}_{lasso}$. In other words, setting $\alpha$ too strictly will result in dimmer pixels to have $\widehat{\mathbf{w}}_{lasso} = \mathbf{0}$, while setting $\alpha$ too loosely will result in spurious non-zero values in $\widehat{\mathbf{w}}_{lasso}$ for brighter more complex pixels, potentially not passing the elbow filter and thus $\widehat{\mathbf{w}}_{OLS} = \mathbf{0}$. To accommodate a wide range of rolony intensities, we choose $\alpha$ adaptively based on the pixel norm $||\mathbf{y}||_2$. First, we form a training data from a random subset of foreground pixels indexed by $i$. For a given pixel norm $u$, we find the alpha that maximizes a weighted sum of $||\widehat{\mathbf{w}}^{(i)}_{OLS}||_1$ giving more weights to training pixels with closer norms to $u$ (equation $*$):

$$\alpha(u) = argmax_\alpha \sum_i g\left(\frac{u - ||\mathbf{y}^{(i)}||_2}{\sigma}\right)||\widehat{\mathbf{w}}^{(i)}_{OLS}(\alpha)||_1$$

where $g(.)$ is the Gaussian function. In practice, for the training pixels we solve the sparse decoding problem for every value of $\alpha$ on a grid from 0.01 to 0.1 with a step size of 0.005, $\boldsymbol{\alpha}_{train}$, to obtain estimated weights $\widehat{\mathbf{w}}^{(i)}_{OLS}(\alpha)$. Then we create a grid of norms $\boldsymbol{u}_{train}$, spanning 0 and 2.8 with 50 steps. For every value of $u$ in $\boldsymbol{u}_{train}$, we solve equation $*$ on the $\boldsymbol{\alpha}_{train}$ grid. In other words, we create a lookup table connecting values of $\boldsymbol{u}_{train}$ to the best $\alpha$ in $\boldsymbol{\alpha}_{train}$. For new pixels, $\alpha$ is determined by the closest norm in the lookup table.

**Spot calling.** To call spots, Gaussian smoothing is applied to individual OLS maps, followed by *peak_local_max* filter (scikit-image 0.19.3[91]) which returns a binary image with 1's at the local maxima of the smoothed OLS maps. These peaks are then used as markers for watershed segmentation. From each segmented region, the following features are retained to be used in downstream steps: area, centroid, maximum and average intensity. This formed a list of candidate spots from each FOV.

**Spot filtering.** To control the specificity of the decoding procedure, we augmented the codebook with a number of barcodes (5-10% of the used barcodes) not used in the probe set (empty barcodes). After spot calling, we record the properties (e.g., area, maximum and average intensity) of spots with an empty barcode. Indeed, we see that empty spots tend to be smaller with lower average/maximum weight (Supplementary Fig. 2c and d). On a small fraction of spots from all fields of view, we train a random forest classifier (scikit-learn v1.1.3) with area, maximum and average weights as features to predict empty/non-empty labels (Supplementary Fig. 2e). We applied the classifier to all spots and obtained emptiness probabilities and set a threshold on these probabilities (0.3–0.35).

**Spot assignment to cells.** The cell boundaries were computed by applying *find_boundaries* (scikit-image 0.19.3[91]) to the segmentation mask. The distances of all spots were calculated to the closest boundary pixel. The distance was set to 0 if a spot was inside a boundary. A spot was assigned to its closest cell if the distance was less than or equal to 11µm in the kidney, 3µm for non-*MBP* and 0µm for MBP spots in the brain.

**Cell annotation**
We used anndata[92–94] (v0.8.0) and scanpy[92](v1.9.1) to handle and analyze the data. The data normalization was performed using analytic Pearson residuals[95] (clipped at 40) with a lower bound placed on gene-level standard deviations[96]. Clustering was done with the Leiden algorithm[97] implemented in scanpy.

**Annotating the Brain data set.** Cells with counts less than 5 and more than 300 were removed (2980 out of 26348). The top 100 highly variable genes (*scanpy.experimental.pp.highly_variable_gene(.,*

*flavor='pearson_residuals')*) were used for normalization, embedding and annotations. PCA was performed on pearson residuals, and the neighborhood graph was created with this command *scanpy.pp.-neighbors(., n_neighbors = 20, n_pcs = 15, metric='cosine')*. Single-nucleus RNA-seq reference from Jorstad et al.[61] was subsetted to M1C cells and normalized in the same way as DART-FISH. Pax6 and Scng subclasses were removed since we did not design our probe set to target those. Average normalized counts (centroids) were computed for every other subclass in the "*within_area_subclass*" slot and all clusters of DART-FISH. To annotate the DART-FISH clusters at the class level (excitatory, inhibitory, non-neuronal), we first correlated each cluster to all single-nucleus subclasses, and assigned that cluster to the class of the most highly correlated subclass. Annotation of each class was done separately.

For excitatory neurons, all DART-FISH cells that had a class label of "excitatory" and had at least 20 transcripts were kept (5957 cells). We realized that the Leiden clustering was unstable and by mere shuffling of the order of cells, we would obtain very different clusters. We reasoned that by removing some cells that tend to move between clusters, we could get more stable clusters and have more confidence in their annotation. To find cells that don't stably cluster, we ran clustering 20 times, every time shuffling the order of the cells. For every cell, we calculated the number of times it was co-clustered with every other cell and took the average of the non-zero values as the co-clustering index (CCI). A perfect CCI of 20 means that the cell is clustered with the same partners in every clustering instance, while lower values show deviations from this limit. We removed the cells with a CCI smaller than 6 and repeated this filtering procedure for three more iterations. The final results show a more stable clustering of the remaining 5101 cells. We then constructed a new neighborhood graph using newly computed principal components (n_neighbors=10, n_pcs=15), followed by Leiden clustering. The cluster centroids were calculated and correlated to the reference subclass centroids. We assigned clusters to their maximally correlated reference subclass if we could also see differential expression of their marker genes (scanpy's *rank_genes_groups*), otherwise we labeled them as NA. Of note, the DART-FISH population labeled as L6b/CT was highly correlated with reference subclasses L6b and L6 CT (Supplementary Fig. 5b) and showed expression of marker genes from both subclasses.

For inhibitory neurons and non-neuronal cells, the clustering was more stable to begin with, and we started by constructing the neighborhood matrix (For inhibitory neurons: n_neighbors=20, n_pcs=10. For non-neuronal cells: n_neighbors=25, n_pcs=15) and clustering. Then clusters were assigned to the reference subclass with maximum Pearson's correlation if the marker genes matched, or otherwise were labeled as NA.

**Drawing cortical layer boundaries.** Cortical layer boundaries were automatically drawn via Support Vector Machine (SVM) decision boundaries. The Scikit-learn python package (v1.1.3) was used to train a SVM on the following excitatory neuron subtype labels: "L2/3 IT", "L4 IT", "L5 IT", "L6 IT", "L6b/CT". First, cells with fewer than 10 total gene counts were filtered out. The x and y coordinates of the cells are standardized via the *StandardScaler()* function, and the data was fed into a SVM with a radial basis function (RBF) kernel with balanced class weights and one vs. one decision function. The RBF SVM model is then applied to a meshgrid with a fine step size with the same geometric size as the original tissue image. The trained SVM classified the cell type label of each point on the meshgrid to define borders between the cortical layers specified by the excitatory neuron subclasses. We drew contours based on the borders between the various subclasses, and manually superimposed them onto Fig. 3c.

**Gene concordance analysis.** The RNA portion of the SNARE-seq2 (snare) dataset from Bakken et al.[46] and Plongthongkum et al.[50] was

used in this section. First, the snare data was subsetted to the DART-FISH genes. Then, DART-FISH and snare data were both normalized (*scanpy.pp.normalize_total(., target_sum = 1000)*) followed by log-normalization (*scanpy.pp.log1p(.)*). The average normalized gene expression was calculated for all subclasses. For each gene, the concordance was defined as the Pearson's correlation between the average expressions across the subclasses between the DART-FISH and snare data (top panel of Supplementary Fig. 5c). The same analysis was performed for a MERFISH data set from Fang et al.[59] (sample H18.06.006.MTG.250.expand.rep1) with the following details: the subclass labels from metadata column "cluster_L2" were renamed to be consistent with DART-FISH annotations. In particular, subclasses L6b and L6 CT were merged, and subclass L5 ET was removed. Note that subclasses Sst Chodl, Chandelier and Lamp5 Lhx6 were not annotated in the MERFISH dataset and were removed from the DART-FISH analysis for consistency. The rest of the analysis was carried out with 242 shared genes between the datasets (bottom panel of Supplementary Fig. 5c).

**Annotating the kidney data set.** Cells with less than 5 and more than 100 transcripts were filtered (2024 out of 65565). The top 250 highly variable genes were kept for downstream analyses (*scanpy.experimental.pp.highly_variable_gene(., flavor='pearson_residuals')*). PCA was performed on pearson residuals, and the neighborhood graph was constructed using the command *scanpy.pp.neighbors(., n_neighbors = 20, n_pcs = 20, metric='cosine')* followed by Leiden clustering (l1 clustering). From the kidney reference atlas[73], degenerative, cycling, transitioning and medullary cell types were removed. The counts were transformed to pearson residuals and the remaining subclass level 1 and level 2 centroids were calculated. We then calculated the Pearson correlations between subclass level 1 centroids and cluster centroids and assigned each l1 cluster to the subclass level 1 with maximum correlation. We then subclustered each of the l1 clusters and assigned those to subclass level 2 identities with maximum correlation, only if the relevant marker genes were expressed. Through this procedure we could not resolve PT-S1 and PT-S2 subtypes separately; thus, we labeled the clusters that were highly correlated with these populations as PT-S1/S2. Similarly, for immune cells, this procedure could confidently resolve MAC-M2 cells and the general myeloid (IMM_Myl) and lymphoid (IMM_Lym) populations. To annotate the immune cells at higher level of granularity, we updated their subclass level 2 labels with the following strategy: Each DART-FISH cell with subclass level 1 label "IMM" was separately correlated with the following immune subtypes in the reference atlas: B, PL, T, MAC-M2, MDC, cDC. The immune subtypes with highest and 2nd highest correlation were kept. If the highest correlation was larger than 0.4 and the ratio of the highest to the 2nd highest correlation was larger than 1.25, the label was updated to that of the highest correlated subtype, otherwise it remained unchanged.

**Cell-cell interaction analysis.** We used *squidpy.gr.co_occurrence* function (v1.2.4.dev27+gb644428) with *n_splits = 1* and an interval between 7μm and 110μm[77].

**Comparison of decoding methods.** Datasets of varying levels of complexity were simulated to compare SpD with StarFish[42] (pixel-based naive matching), BarDensr[44] and ISTDECO[43] (deconvolution-based methods). The synthetic datasets were constructed using the human brain codebook (3-on-3-off, 121 genes with 10 empty barcodes) with equal abundance of all genes and uniform spatial distribution of spots. The rolonies were modeled as gaussian spots with peak intensity randomly chosen to be between 0.25 and 0.7 and sigma between 2 and 2.5 pixels. To model channel-specific intensity variation, we randomly drew 18 channel-specific coefficients from a uniform distribution between 0.75 and 1.25 to scale their respective images, while clipping the intensity values above 1. We simulated multiple datasets varying the number of spots between $5*10^3$ to $4*10^5$ spots in a field of view of size 1024 x 1024 pixels. Different decoding methods were applied to the synthetic datasets with default settings to the extent possible, with no post-hoc filtering of the spots. The only exception was StarFish for which the distance threshold was set to 0.7 as a fair balance between specificity and sensitivity. Then, the groundtruth spots were matched one-to-one to the decoded spots if the barcodes were identical and the centroids were closer than 6 pixels. Sensitivity is defined as the fraction of groundtruth spots matched with a decoded spot. Specificity is defined as the fraction of matched decoded spots over all decoded spots. Empty rate is the fraction of empty barcodes among all decoded barcodes and is inversely related to specificity.

## Reporting summary
Further information on research design is available in the Nature Portfolio Reporting Summary linked to this article.

## Data availability
The spot tables, RiboSoma images and segmentation masks are available on figshare for human brain (https://doi.org/10.6084/m9.figshare.23932863.v1)[98] and for human kidney (https://doi.org/10.6084/m9.figshare.23937057.v1)[99]. All registered DART-FISH images, codes and intermediate outputs of the processing pipeline are available on Zenodo (https://doi.org/10.5281/ZENODO.8253771)[100]. Source data are provided with this paper. The single-nucleus RNA sequencing reference atlas of human kidney[73] is available on GEO (GSE183277). SNARE-seq data for human M1C[46,50] is available at Brain Cell Data Center (https://biccn.org/data) under U01 ZhangKun grant ID (U01MH114828). The M1C data from Jorstad et al.[61] is available for download from the Neuroscience Multi-omics Archive (https://data.nemoarchive.org/publication_release/Human_Cross_Areal_Analysis/). Source data are provided with this paper.

## Code availability
The python code for the DART-FISH processing pipeline and SpD are available on this Github repository: https://github.com/Kiiaan/DF3D.

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

## Acknowledgements

The authors would like to thank Drs. Prashant Mali, Reza Kalhor, Van Ninh, Eric Griffis, Xiaohua Huang and Bing Ren for useful discussions and comments on this work. The authors would like to acknowledge Kimberly Conklin, Huy Lam and other members of the Zhang lab for their support. This work was supported by NIH grants U01MH098977 (to K.Z., J.C., J.F.), U01MH114828 (to K.Z., J.C., P.V.K.), UG3/UH3DK114933 (to K.Z., S.J.), U54HL145608 (to K.Z., S.J., P.V.K.), R01AG065541 (to J.C.).

## Author contributions

K.Z. and J.B.F. conceived the in situ decoding concept. C.J.C. performed human brain experiments. K.K. performed human kidney experiments, created the data processing pipeline and analyzed the data. K.K., C.J.C. and M.N. optimized the protocol on human tissues. H.S.L., M.C. and R.Q. performed early developments. C.D.K. and E.L. contributed human brain sections. C.R.P. and J.C. prepared human brain sections and helped with interpretation of the brain data. Y.Y. performed computational layer boundary detection in the human brain. J.S. provided key suggestions on protocol optimization and cell type annotation of the human brain. X.L. assisted in the mouse kidney experiment. Y.Z. and P.V.K. performed gene selection for human kidney with input from B.B.L. and S.J. A.K., J.P.G. and S.J. contributed human kidney sections, performed histology and review. S.J. and B.B.L. helped with interpretation of the kidney data. K.K., C.J.C. and K.Z. wrote the manuscript with suggestions from all authors. K.Z. supervised the project.

## Competing interests

K.Z. and J.B.F. are listed as inventors in a patent related to the method described in this manuscript. All remaining authors declare no competing interests.
