## [Peer Review File · Nature Communications]

Mapping human tissues with highly multiplexed RNA in situ hybridizationREVIEWERS' COMMENTS

Reviewer #1 (Remarks to the Author)

Thank you to the authors for submitting this manuscript describing DART-FISH as an in situ multiplexed approach to RNA mapping in tissues. It is well written, thoughtful and builds constructively upon previous work whilst additionally demonstrating application in a variety of scenarios of varying degrees of challenge (healthy brain, diseased kidney). The figures are appropriate and impactful. The approach has a place in the evolving landscape of spatially empowered RNA localisation technologies. My only query/request is a minor one:

1) The one figure/table that would add additional but appropriate impact would be a representation of the key characteristics of this approach (including pros/cons & objective points such as resolution/expense/surface area/optimisation/etc) compared to other various spatial technologies.

Reviewer #2 (Remarks to the Author)

The paper describes a method for in situ transcriptomics by padlock probing, rolling circle amplification, and an image-based approach for combinatorial labelling followed by decoding the resulting image data by deconvolution. The paper also presents application of the method to several samples of human tissue. As a whole, the paper is well written and methods are clearly described, but the authors seem unaware of the development of the field over the past ten years; the presented methods lack novelty and references to the field are missing.

First of all, the presented DART-FISH approach is very similar to the in situ RNA sequencing presented in Nature methods in 2013 by Ke et al: <https://www.nature.com/articles/nmeth.2563> This method has also been refined and is now available as the Xenium In Situ approach by 10X genomics, https://pages.10xgenomics.com/tch-2023-04-tech-lit-ra_g-p_xenium-performance-data-lp.html

Furthermore, the 'omni-cell type cytoplasmic stain RiboSoma' is the same thing as the anchor probe in the above mentioned paper, and such anchor probe signals were previously used for cell segmentation, e.g. in a Nature Methods publication from 2021 by Park et al: <https://www.nature.com/articles/s41467-021-23807-4>

Finally, the 'decoding by deconvolution', described in the paper seems to be a variant of the code-book-based decoding presented in PLOS computational Biology by Chen et al in 2021: <https://journals.plos.org/ploscompbiol/article?id=10.1371/journal.pcbi.1008256> Note that this paper is included as reference 74 in the paper, saying that 'more sophisticated deconvolution methods that share information between neighboring pixels may improve decoding efficiency', which is true, and one wonders why open code from previous publications has not been considered or benchmarked on the presented data.

It is worth noting that the 'proof-of-principle' results are interesting, but there is no noteworthy novelty in the methods presented. This makes the methods of less interest for the field – especially since no code or data is openly shared, meaning that it will be difficult for others to apply the decoding approach to own data. Plans to share code and data are mentioned, but at the time of review nothing is available, also making it difficult to judge the usability of the code.

The work supports the conclusions in terms of being functioning and potentially useful, but the work lacks novelty. Additional comparisons to above mentioned methods and their developments would be needed to claim that the proposed methods are an improvement of the state-of-the-art.

In-depth publications on the results from the findings of the application of the method, including follow-up experiments, would likely be of interest for the community.

Reviewer #3 (Remarks to the Author)

Kalhor et al. have introduced a novel in situ transcriptomic technique called DART-FISH, which utilizes padlock probes and rolling circle amplification to detect RNA species in their spatial context. The authors have provided comprehensive protocols for sample preparation, probe design, and imaging, facilitating replication of the technique. Moreover, they have demonstrated the high sensitivity and specificity of DART-FISH by analyzing data from various human tissues, showcasing its potential in mapping cell types, identifying gene expression patterns, and studying cell-cell interactions in pathological niches. The manuscript also includes a comparison of DART-FISH with other RNA in situ hybridization methods and discusses its applications in neuroscience, cancer, and developmental biology research. Overall, this work provides valuable insights into the latest techniques for mapping human tissues using RNA in situ hybridization.

However, I have a few major concerns regarding the manuscript:

1. While the major procedures of DART-FISH share similarities with other in situ hybridization methods, it would be beneficial for the authors to highlight the unique advantages of DART-FISH more explicitly in addition to array synthesis of padlock probes and the deconvolution of fluorescent signals.
2. Although the authors briefly discuss the potential for false positives due to cross-reactivity, they have not conducted a detailed analysis of the factors contributing to this phenomenon. Addressing this limitation in future work would be crucial to fully understand and mitigate false positives in DART-FISH.
3. The reproducibility of DART-FISH has been demonstrated by Kalhor et al., but they have not thoroughly investigated the factors that contribute to result variability. This limitation should be acknowledged and further explored, particularly for samples that may be more prone to variability.
4. It is important to note that the DART-FISH technique is applicable only to fresh-frozen tissue sections. However, the manuscript does not evaluate the effects of different fixation methods, processing techniques, and RNA degradation on the results. Assessing these factors would provide a more comprehensive understanding of the limitations and potential improvements of the technique.
5. The incubation time, efficiency of stripping and washing away the unbound decoding probes are crucial factors that can potentially impact the quality of the obtained transcriptomic data. Therefore, it is imperative to address the variations in these factors in this work. By investigating and reporting on these variables, the study can provide a comprehensive understanding of their influence on the experimental outcomes. This will enhance the reliability and robustness of the technique.

RESPONSE TO REVIEWERS' COMMENTS

In response to the constructive critiques made by the three reviewers, we have added results from new experiments and analyses, and revised the manuscript to clarify a few key points. The major changes are summarized below.

1. We added a **Supplementary Table S1** to compare DART-FISH with other published RNA in situ methods
2. In **Supplementary Fig. S1d** we added results of new experiments showing that colonies are stable through multiple rounds of imaging, the signal levels do not dwindle and the background level does not rise.
3. In **Supplementary Fig. S2f**, we benchmarked SpD with multiple published methods. We showed that SpD outperforms naive decoding methods currently widely used in the field.
4. In the discussion section, per reviewer #3's request, we outlined our deliberations about tissue quality, fixation and pretreatment. We proposed RiboSoma as a readout that enables fast comparison between different tissue conditions.
5. We edited the introduction and discussion sections of the manuscript to reflect reviewers' requests.
6. We have shared the protocol, codes and data through protocols.io, Github, Zenodo and Figshare.

Reviewer #1 (Remarks to the Author)

Thank you to the authors for submitting this manuscript describing DART-FISH as an in situ multiplexed approach to RNA mapping in tissues. It is well written, thoughtful and builds constructively upon previous work whilst additionally demonstrating application in a variety of scenarios of varying degrees of challenge (healthy brain, diseased kidney). The figures are appropriate and impactful. The approach has a place in the evolving landscape of spatially empowered RNA localisation technologies. My only query/request is a minor one:

Author response: We thank this reviewer for recognizing DART-FISH, in particular our efforts on demonstrating it using more challenging tissue types.

1) The one figure/table that would add additional but appropriate impact would be a representation of the key characteristics of this approach (including pros/cons & objective points such as resolution/expense/surface area/optimisation/etc) compared to other various spatial technologies.

Author response:

We agree that a comparison with existing methods in this field is helpful and have added a **Supplementary Table S1**. There are however caveats that can make the comparison tricky. For example, the cost would depend on the number of genes to target, how to produce the probes, how many experiments each probe set will be used for etc. We discussed these complexities in the footnotes.

Technology	#genes	#decoding cycles	barcode space used ¹	enzymatic signal amp.	Objective magnification ²	#probes per gene	Probe set cost for 300 genes ⁴	between cycle preparation time	cell segmentation	tissues tested
DART-FISH	120-300	7	22-31%	Yes	20x	50	\$5200 (Twist Bio)	45 minutes	RiboSoma, nuclear	Human brain, Human Kidney
ISS (Qian et al 2020)	99	5	15% ³	Yes	20x	7-8	\$136,000 (IDT 4nm Ultramer plate with 5' phosphorylation)	>1.5 hours	nuclear	Mouse Brain
HybISS (Gyllborg et al 2020)	120	5	11% ³	Yes	40x	5	\$52,000 (IDT 4nm Ultramer plate without 5' phosphorylation)	3.5 hours	nuclear	Human brain, Mouse brain
STARmap (Wang et al 2018)	160-1000	6	15-97% ³	Yes	40x	4	\$38,000 (IDT plates with 5' phosphorylation)	4.5 hours	nuclear	Mouse brain
MERFISH (Fang et al 2022)	250-4000	16	6-96%	No	60x	60	\$5200 (Twist Bio)	35 minutes	Poly dT, nuclear	Human brain, Mouse brain

¹ Number of used barcodes over number of valid barcodes. Lower ratios enable better error detection and signal demixing.

² Higher magnification is required for smaller and dimmer features. On the other hand, lower magnification objectives can image larger areas faster.

³ 4-color imaging was used instead of 3-color imaging

⁴ Cost of probe sets cannot be accurately compared across technologies for the following reasons: 1) Each technology uses a different number of probes per gene. For example, smFISH-based technologies (e.g., MERFISH) need a minimum number of probes for the signal to be detectable, while padlock-probe based technologies enjoy higher sensitivity with more padlocks per gene. Cost of direct synthesis scales linearly with the number of probes, thus technologies that use direct synthesis tend to keep the number of probes per gene 3) each technology may use a different mass of probes per experiment which makes it very difficult to estimate cost per experiment. We tried to estimate the cost of purchasing the probe sets with the best knowledge obtained from the manuscripts. These values do not include the costs associated with the preparation of these probes for an experiment (e.g., amplification, phosphorylation)

Reviewer #2 (Remarks to the Author)

The paper describes a method for in situ transcriptomics by padlock probing, rolling circle amplification, and an image-based approach for combinatorial labelling followed by decoding the resulting image data by deconvolution. The paper also presents application of the method to several samples of human tissue. As a whole, the paper is well written and methods are clearly described, but the authors seem unaware of the development of the field over the past ten years; the presented methods lack novelty and references to the field are missing.

Author response: We thank this reviewer for the positive comments on our method and the writing. We have cited the relevant publications, and added additional clarifications in order to place this work in a bigger context (see below).

First of all, the presented DART-FISH approach is very similar to the in situ RNA sequencing presented in Nature methods in 2013 by Ke et al: <https://www.nature.com/articles/nmeth.2563> This method has also been refined and is now available as the Xenium In Situ approach by 10X genomics, https://pages.10xgenomics.com/tch-2023-04-tech-lit-ra_g-p_xenium-performance-data-lp.html

Author response: Citation to the foundational work by Ke et al (2013) has been added to the manuscript. Our method is indeed conceptually similar to Ke et al in that it uses RCA to amplify barcoded padlock probes. But we have major differences in the synthesis of the probes, cDNA preservation and decoding. Specifically,

1. Our padlock probe production strategy vastly reduces the cost and allows for targeting of more gene while increasing the number of probes per gene (**Supplementary Fig. S3**),
2. We showed that there is cDNA loss without thorough crosslinking of the molecules. Therefore, we modified the chemistry such that the cDNA molecules become a part of a polyacrylamide gel after reverse-transcription (**Supplementary Fig. S1a**). Better preservation of the cDNA led to higher density of colonies (**Supplementary Fig. S1b-c**)
3. The decoding method implemented in DART-FISH is the fastest and the most robust among all the current in situ transcriptomic technologies. In DART-FISH, short oligonucleotides hybridize to colonies, hence it enjoys both a high SNR conferred by the RCA and fast hybridization and stripping kinetic.

Furthermore, the ‘omni-cell type cytoplasmic stain RiboSoma’ is the same thing as the anchor probe in the above mentioned paper, and such anchor probe signals were previously used for cell segmentation, e.g. in a Nature Methods publication from 2021 by Park et al: <https://www.nature.com/articles/s41467-021-23807-4>

Author response: What the reviewer is describing as “anchor probe signals” in Ke et al 2013, is a stain that targets all colonies only from the genes of interest simultaneously. The anchor primer is a sequence that is shared between all padlock probes. Indeed, we have a similar sequence, i.e., the RCA primer binding site, and call it by the same name. RiboSoma on the other hand, is a stain for all cDNA molecules from all transcripts present in the tissue, not just colonies. We designed our random nonamer (N9) and poly-dT reverse-transcription primers to have a constant 20-nucleotide sequence on their 5’ end, which enables their targeting with fluorescent oligos. We call this cDNA stain RiboSoma because it lights up the cytoplasm as well as nuclei. Note that, the goal in Park et al 2021 “Cell segmentation-free inference of cell types from in situ transcriptomics data” is not to perform cell segmentation, but to skip this step and assign cell types to regions in the space with dense transcript counts.

Finally, the ‘decoding by deconvolution’, described in the paper seems to be a variant of the code-book-based decoding presented in PLOS computational Biology by Chen et al in 2021: <https://journals.plos.org/ploscompbiol/article?id=10.1371/journal.pcbi.1008256> Note that this paper is included as reference 74 in the paper, saying that ‘more sophisticated deconvolution methods that share information between neighboring pixels may improve decoding efficiency’, which is true, and one wonders why open code from previous publications has not been considered or benchmarked on the presented data.

Author response: We thank the reviewer for this suggestion. To benchmark different methods, we generated simulated data with ground truth and compared SpD with the method from Chen et al (2021). The results of this comparison are shown in **Supplementary Fig. S2f**.

It is worth noting that the 'proof-of-principle' results are interesting, but there is no noteworthy novelty in the methods presented. This makes the methods of less interest for the field – especially since no code or data is openly shared, meaning that it will be difficult for others to apply the decoding approach to own data. Plans to share code and data are mentioned, but at the time of review nothing is available, also making it difficult to judge the usability of the code.

Author response: Since the time of our first submission, we have made all our data and codes available. The code for the processing pipeline resides here: <https://github.com/Kiiaan/DF3D>
The raw images, the output of the pipeline alongside the codes used for its generation are here: <https://doi.org/10.5281/ZENODO.8253771>

Since the images are large, we also uploaded spot tables, segmentation masks and RiboSoma images in a separate repository for easier access:

<https://figshare.com/s/5486fc7a7465acf38a63> (brain) and
<https://figshare.com/s/1df969c5fdc81127956a> (kidney)

The work supports the conclusions in terms of being functioning and potentially useful, but the work lacks novelty. Additional comparisons to above mentioned methods and their developments would be needed to claim that the proposed methods are an improvement of the state-of-the-art.

Author response: The main message of our manuscript is that we developed an in situ RNA mapping method that employs a simple decoding procedure, has a built-in cytoplasmic stain, is not limited by the length of the target gene, and can generate meaningful data from fresh frozen human tissues. While the overall concept looks similar to several other published methods, the individual components in this study are novel. We provided plenty of discussions and examples to support each part of this message. For example, 1) we showed that our cytoplasmic stain can help with protocol optimization through cDNA retention as well as aiding with cell segmentation in two very different tissue types. 2) Our human kidney data set to our knowledge is the first high-throughput in situ data available for this tissue. 3) In the human brain, we showed how targeting short neuropeptides can uncover very rare cell types.

In-depth publications on the results from the findings of the application of the method, including follow-up experiments, would likely be of interest for the community.

Author response: As a methods paper, we focused the description on our approach, compared it with the current state of the field, and presented its application and its unique capabilities. We are currently using DART-FISH in our studies of kidney diseases, as well as the human brain mapping effort under the NIH BRAIN initiative. The results of these studies will be published in dedicated and biology-focused manuscripts.

Reviewer #3 (Remarks to the Author)

Kalhor et al. have introduced a novel in situ transcriptomic technique called DART-FISH, which utilizes padlock probes and rolling circle amplification to detect RNA species in their spatial context. The authors have provided comprehensive protocols for sample preparation, probe design, and imaging, facilitating replication of the technique. Moreover, they have demonstrated the high sensitivity and specificity of DART-FISH by analyzing data from various human tissues, showcasing its potential in mapping cell types, identifying gene expression patterns, and studying cell-cell interactions in pathological niches. The manuscript also includes a comparison of DART-FISH with other RNA in situ hybridization methods and discusses its applications in neuroscience, cancer, and developmental biology research. Overall, this work provides valuable insights into the latest techniques for mapping human tissues using RNA in situ hybridization.

However, I have a few major concerns regarding the manuscript:

1. While the major procedures of DART-FISH share similarities with other in situ hybridization methods, it would be beneficial for the authors to highlight the unique advantages of DART-FISH more explicitly in addition to array synthesis of padlock probes and the deconvolution of fluorescent signals.

Author response: We provided a new **Supplementary Table S1** in which we performed a comparison between a number of other methods.

2. Although the authors briefly discuss the potential for false positives due to cross-reactivity, they have not conducted a detailed analysis of the factors contributing to this phenomenon. Addressing this limitation in future work would be crucial to fully understand and mitigate false positives in DART-FISH.

Author response: Since the decoding probes are derived from Illumina bead array technologies (Gunderson et al. 2004), they have been thoroughly tested and optimized by Illumina and we expect negligible cross-reactivity between them. As for the specificity of padlock probes, in our early testings we performed experiments with multiple negative-control padlock probes (ones that target genes from other species) and never found non-specific colonies from these probes. We attribute this to the high temperature at which the hybridization/ligation step is performed. The main source of non-specificity usually comes from computational decoding, where due to overcrowding, spots are misassigned to wrong barcodes. As shown in the new **Supplementary Fig. S2f**, this is a common problem amongst all tested decoding algorithms. The best way to solve this problem is to increase the redundancy in the barcode space which leads to longer imaging time and larger data size. Nevertheless at a given level of redundancy, one can mitigate this issue computationally. In the right panel of **Supplementary Fig. S2f**, we demonstrate a relationship between empty rate (fraction of spots assigned to unused barcodes) and specificity. Note that empty rate is an observed variable from an experiment and can be used to control specificity. By keeping a low empty rate (the exact value can be calculated based on the properties of the codebook) one can obtain a high confidence in the specificity of the data. Indeed, in the data shown in **Fig. 3** and **Fig. 4** of the paper, we controlled the empty rate to be lower than 0.25% when 5-8% of the codebook consisted of unused barcodes. We have edited the Results and Discussion sections to communicate these matters more clearly.

3. The reproducibility of DART-FISH has been demonstrated by Kalhor et al., but they have not thoroughly investigated the factors that contribute to result variability. This limitation should be acknowledged and further explored, particularly for samples that may be more prone to variability.

Author response: We believe that tissue quality and pretreatment are the most critical sources of variation in DART-FISH experiments. In a new paragraph in the Discussion section, we acknowledged this limitation and advocated for dedicated surveys of tissue processing and pretreatment to find the best methods suited for each human tissue type.

4. It is important to note that the DART-FISH technique is applicable only to fresh-frozen tissue sections. However, the manuscript does not evaluate the effects of different fixation methods, processing techniques, and RNA degradation on the results. Assessing these factors would provide a more comprehensive understanding of the limitations and potential improvements of the technique.

Author response: As stated above, in the new paragraph, we discuss these limitations. In particular, we discussed the need for optimization of pre-reverse-transcription steps for new tissue types and fixation methods. We proposed that RiboSoma can be a helpful guide in these optimizations.

5. The incubation time, efficiency of stripping and washing away the unbound decoding probes are crucial factors that can potentially impact the quality of the obtained transcriptomic data. Therefore, it is imperative to address the variations in these factors in this work. By investigating and reporting on these variables, the study can provide a comprehensive understanding of their influence on the experimental outcomes. This will enhance the reliability and robustness of the technique.

Author response: We assume that this reviewer is asking about the efficiency of stripping the signal from one round before staining the next round. Indeed, signal carry-over from one cycle to the next will be detrimental to the decoding of colonies. We take extreme care in making sure that the signal is fully stripped before moving forward to the subsequent rounds. Since all staining, stripping and washing steps are performed at room temperature, we could perform these actions as the sample is mounted on the microscope and hence monitor the signal after stripping and washing. The parameters mentioned in the methods section of the manuscript, including incubation times and wash buffer compositions, are the result of excessive caution in stripping the signal. Looking forward, if a user is interested in automating this process, a small subroutine could be developed to check the background levels after stripping and repeat the stripping procedure in case the background signal is higher than expected levels. To fully address the reviewer's concern, we conducted a small experiment on mouse kidney with a handful of probes targeting a few genes. We then stained and stripped the sample several times and recorded images after every step. In the new **Supplementary Fig. S1d**, we add a panel that shows that following the procedure in the methods section, the signal from every round is fully cleared after the stripping procedure. Furthermore, the colonies are stable and the decoding process causes negligible degradation.

REVIEWERS' COMMENTS

Reviewer #1 (Remarks to the Author):

Thank you to the authors for submitting this revised manuscript describing DART-FISH as an in situ multiplexed approach to RNA mapping in tissues. It is well written, thoughtful and builds constructively upon previous work whilst additionally demonstrating application in a variety of scenarios of varying degrees of challenge (healthy brain, diseased kidney). The figures are appropriate and impactful. The approach has a place in the evolving landscape of spatially empowered RNA localisation technologies. All of my previous queries have been adequately addressed and I do not have more at this time.

Reviewer #2 (Remarks to the Author):

The authors have responded to all the review comments, clarifying the novelties in their work. The sharing of data and code is very much appreciated, and increases the value of the work, making publication valuable to the community.

Reviewer #3 (Remarks to the Author):

The authors have resolved all my concerns.